# Hybrid Regret Bounds for Combinatorial Semi-Bandits and Adversarial Linear Bandits

**Shinji Ito**
NEC Corporation
i-shinji@nec.com

## Abstract

This study aims to develop bandit algorithms that automatically exploit tendencies of certain environments to improve performance, without any prior knowledge regarding the environments. We first propose an algorithm for combinatorial semi-bandits with a hybrid regret bound that includes two main features: a best-of-three-worlds guarantee and multiple data-dependent regret bounds. The former means that the algorithm will work nearly optimally in all environments in an adversarial setting, a stochastic setting, or a stochastic setting with adversarial corruptions. The latter implies that, even if the environment is far from exhibiting stochastic behavior, the algorithm will perform better as long as the environment is "easy" in terms of certain metrics. The metrics w.r.t. the easiness referred to in this paper include cumulative loss for optimal actions, total quadratic variation of losses, and path-length of a loss sequence. We also show hybrid data-dependent regret bounds for adversarial linear bandits, which include a first path-length regret bound that is tight up to logarithmic factors.

## 1 Introduction

In this work, we consider two fundamental problem settings w.r.t. online decision problems: *combinatorial semi-bandits* [42, 66, 4] and *linear bandits* [13, 17, 30]. In both problem settings, a player is given an action set $\mathcal{A} \in \mathbb{R}^d$, a compact subset of a $d$-dimensional vector space. In each round $t$, the player chooses an action $a_t \in \mathcal{A}$ and then incurs loss $\ell_t^\top a_t$, where $\ell_t \in \mathbb{R}^d$ is a *loss vector* chosen by the environment. The action set in the combinatorial semi-bandit is assumed to be a subset of $\{0, 1\}^d$, each element of which corresponds to a subset of $[d] = \{1, 2, \ldots, d\}$. After choosing $a_t \in \mathcal{A}$, the player can observe $\ell_{ti}$ for all $i$ such that $a_{ti} = 1$ in semi-bandits. Linear bandits are problems with even more limited feedback, ones in which the learner can only observe the incurred loss $\ell_t^\top a_t$. For combinatorial semi-bandits problems with $\ell_t \in [0, 1]^d$, there is a known algorithm with an $O(\sqrt{mdT})$-regret bound [4], where $m = \max_{a \in \mathcal{A}} \|a\|_1$. For linear bandits such that $|\ell_t^\top a| \leq 1$ holds for any $a \in \mathcal{A}$, algorithms with $\tilde{O}(d\sqrt{T})$-regret (where factors in $\log T$ and $\log d$ are ignored) have been developed [13, 17, 30]. These algorithms are optimal in terms of worst-case analysis. In fact, matching lower bounds of $\Omega(\sqrt{mdT})$ for combinatorial semi-bandits [3, 11] and of $\Omega(d\sqrt{T})$ for linear bandits [22] are known.

The worst-case optimal algorithms, however, tend to be too conservative in actual practice. This is because true worst-case environments are quite rare in real-world applications. Rather, the environments may have structures that are convenient for the learner, and it is desirable that the algorithm takes advantage of such structures to improve performance. To exploit such structures, two main categories of approaches have been studied: adapting to (nearly) stochastic environments and developing data-dependent regret bounds.

An example of the first category is in reference to *best-of-both-worlds (BOBW)* algorithms [12, 62, 5, 61, 68, 70], which means that they work well for both adversarial and stochastic settings.

35th Conference on Neural Information Processing Systems (NeurIPS 2021).

These algorithms enjoy $\tilde{O}(\sqrt{T})$-regret bounds in an adversarial setting and, simultaneously, achieve $O(\frac{(\log T)^c}{\Delta})$-regret in a stochastic setting with i.i.d. losses, where $c \geq 1$ is a constant and $\Delta$ represents the *suboptimality gap* defined by $\min_{a \in \mathcal{A} \setminus a^*} \mathbf{E}[\ell_t^\top (a - a^*)]$ for an optimal action $a^* \in \mathcal{A}$. As shown in Table 1, for combinatorial semi-bandits, Zimmert et al. [71] provide a BOBW algorithm, both bounds of which are tight up to constant factors as matching lower bounds are known for adversarial settings [3, 11] as well as for stochastic settings [42].

Studies on the adversarial robustness of stochastic bandit algorithms [48, 27, 70] can be considered to provide another approach in the first category, in which an adversary can corrupt stochastically-generated losses subject to the constraint that the total amount of the corruption is at most a parameter $C \geq 0$, referred to as the *corruption level*. This model includes both adversarial and stochastic settings and is closely related to studies on BOBW algorithms. In fact, the special cases of $C = 0$ and $C = \Omega(T)$ correspond to stochastic and adversarial settings, respectively. For the fundamental multi-armed bandit problem, Zimmert and Seldin [70] have proposed the Tsallis-INF algorithm, which achieves BOBW with tight regret bounds up to small constant factors, and which simultaneously is very robust w.r.t. corruptions; the degradation in the regret is only $O(\min\{C, \sqrt{\frac{CK \log T}{\Delta}}\})$. Such algorithms are called *best-of-three-worlds (BOTW)* algorithms. As shown in Table 1, the semi-bandit algorithm by Zimmert et al. [71] is also a BOTW algorithm (Sto. +Adv. refers to the stochastic setting with $C$-adversarial corruptions).[1] For linear bandits, Lee et al. [45] have recently developed a BOTW algorithm as well.

In studies on data-dependent regret bounds [55, 68, 29, 57, 15, 14, 34] of the other category, we define measures of the difficulty of problem instances, which we refer to as *difficulty indicators*, and aim to develop algorithms so that the regret will be smaller for smaller instance-difficulty indicators. Examples of difficulty indicators dealt with in this paper include $L^*$: the cumulative losses for an optimal action, $Q_q$: the total quadratic variation of losses, and $V_q$: the path-length of losses, definitions of which are given in Table 3. Tables 1 and 2 show data-dependent regret bounds, including difficulty indicators for, respectively, combinatorial semi-bandits and adversarial linear bandits. For example, the semi-bandit algorithm with an $O(\sqrt{dL^* \log T})$-regret bound given by Wei and Luo [68] achieves much smaller regret than the worst-case optimal $O(\sqrt{dmT})$-bound when $L^* = o(mT)$, i.e., when there exists an action for which the cumulative losses are much smaller than $mT$.

Given the various algorithms above, a new challenge arises: how, in practice, can we choose an appropriate algorithm? If the environment is expected to behave in an almost i.i.d. stochastic manner, either BOBW or BOTW algorithms would work well. If the environment is far from exhibiting stochastic behavior but is "easy" in terms of some difficulty indicator, algorithms with a corresponding data-dependent regret bound may work better. In practice, however, it is hard to tell what kind of environment we are working in until we have actually tried out the algorithm.

## 1.1 Contributions of this work

This work addresses the above-mentioned issue of algorithm selection by developing *hybrid* algorithms. Its main contribution is to develop a semi-bandit algorithm (Algorithm 1) that enjoys multiple data-dependent regret bounds as well. This can be seen as an an extension of the work on the multi-armed bandit problem by Ito [34] to combinatorial semi-bandits. In addition to this, for linear bandits, we provide a hybrid data-dependent regret bound.

For combinatorial semi-bandits problems, we propose Algorithm 1 with the regret bounds shown in Table 1. More explicit statements are provided in Theorem 1, Corollary 1 (for the adversarial setting), and Corollary 2 (for the stochastic setting with/without adversarial corruption). As can be seen in the table, the regret bound for the adversarial setting encompasses three different data-dependent regret bounds. Note that $O(\sqrt{dL^* \log T})$-bounds imply the nearly worst-case optimal bound of $O(\sqrt{dmT \log T})$, as $L^* \leq mT$ always follows from the model definition. The new regret bound is of $O(R + \sqrt{RCm})$ for the corrupted stochastic setting, where $R$ stands for the bound for the stochastic setting. Note that the regret bounds for the stochastic settings (with corruption) can be improved for some special cases, such as for size-invariant ($\|a\|_1 = m$ for all $a \in \mathcal{A}$) or for a matroid constraint ($\mathcal{A}$ forms bases of a matroid), as described in Corollary 2. We would also like to stress that the

---

[1]This bound is not explicitly stated in their paper, but can be shown via a straightforward modification of the proof. A proof of this is given in Appendix B of the supplementary material.

Table 1: Regret bounds for combinatorial semi-bandits.

| Reference | Regime | Regret bound |
|-----------|--------|--------------|
| Audibert et al. [4] | Adv. | $O(\sqrt{dmT})$ |
| Kveton et al. [42] | Sto. | $O\left(\frac{dm\log T}{\Delta}\right)$ |
| Neu [54] | Adv. | $O(m\sqrt{dL^*\log(d/m)})$ |
| Wei and Luo [68] (Sec. 3.1) | Adv. | $O(\sqrt{dQ_2\log T})$ |
| Wei and Luo [68] (Sec. 4.1) | Adv. | $O(\sqrt{dV_1\log T})$ |
| Wei and Luo [68] (Sec. 4.2) | Adv. | $O(\sqrt{dL^*\log T})$ |
| Zimmert et al. [71] | Adv. | $O(\sqrt{dmT})$ |
| | Sto. | $O\left(\frac{dm\log T}{\Delta}\right)$ |
| | Sto. + Adv. | $O\left(\frac{dm\log T}{\Delta} + \sqrt{\frac{Cdm^2\log T}{\Delta}}\right)$ |
| **[This work]** (Algorithm 1) | Adv. | $O(\sqrt{d\min\{L^*, Q_2, V_1\}\log T})$ |
| | Sto. | $O\left(\frac{dm\log T}{\Delta}\right)$ |
| | Sto. + Adv. | $O\left(\frac{dm\log T}{\Delta} + \sqrt{\frac{Cdm^2\log T}{\Delta}}\right)$ |

Table 2: Regret bounds for adversarial linear bandits. We assume that $\|a\|_p \le 1$ for all $\mathcal{A}$ and $\|\ell_t\|_q \le 1$ for all $t$, for some $p, q \in [1, \infty]$ such that $1/p + 1/q = 1$. $\tilde{O}(\cdot)$ ignores factors of $(\log T)^{O(1)}$ and $(\log d)^{O(1)}$.

| Reference | Regret bound |
|-----------|--------------|
| Bubeck et al. [13] | $\tilde{O}(d\sqrt{T})$ |
| Hazan and Kale [29] | $\tilde{O}(d\sqrt{\vartheta Q_2})$ |
| Bubeck et al. [15] (Cor. 4, 8) | $\tilde{O}(d\sqrt{\vartheta V_2})$ |
| Bubeck et al. [15] (Cor. 6) | $\tilde{O}(d^{3/2}\sqrt{\vartheta V_q})$ |
| Ito et al. [36] | $\tilde{O}(d\sqrt{\min\{L^*, Q_q\}})$ |
| **[This work]** | $\tilde{O}(d\sqrt{\min\{L^*, Q_q, V_q\}})$ |

Table 3: Definitions of parameters.

| | |
|---|---|
| $T$ | number of rounds |
| $d$ | dimension of $\mathcal{A}$ |
| $m$ | $\max_{a\in\mathcal{A}}\|a\|_1$ |
| $\vartheta$ | self-concordant parameter |
| $C$ | corruption level |
| $L^*$ | $\min_{a^*}\sum_{t=1}^{T}\ell_t^\top a^*$ |
| $Q_q$ | $\sum_{t=1}^{T}\|\ell_t - \bar\ell\|_q^2$ |
| | $\left(\bar\ell = \frac{1}{T}\sum_{t=1}^{T}\ell_t\right)$ |
| $V_q$ | $\sum_{t=1}^{T-1}\|\ell_t - \ell_{t+1}\|_q$ |
| $\Delta$ | $\min_{a\in\mathcal{A}\setminus\{a^*\}}\mathbf{E}[\ell_t^\top(a - a^*)]$ |

proposed algorithm is parameter-free, i.e., it does not require any prior information w.r.t. parameters $\Delta, L^*, Q_2, V_1$, and $C$.

The proposed semi-bandit algorithm is based on a follow-the-regularized-leader (FTRL) framework [70, 71], combined with an optimistic prediction for the losses [58, 57, 68]. More precisely, it uses a mixture regularizer [14, 55, 71, 24] consisting of the log-barrier in variables $x_i$ and the Shannon entropy in the *complement* $(1 - x_i)$ of $x_i$ with entry-wisely adaptive learning rates, by which BOTW is achieved. The regret analysis for the stochastic setting (with corruption) is based on self-bounding inequalities for the regret, similarly to what is seen in the analyses by Zimmert et al. [71], Zimmert and Seldin [70]. In addition to this, by choosing optimistic predictors with simple gradient descent methods, we can achieve multiple data-dependent regret bounds as well. The most relevant algorithms are given by Wei and Luo [68] and Zimmert et al. [71]. The proposed algorithm differs from that of Wei and Luo [68] in that the former employs follow-the-regularized-leader rather than online mirror descent methods and uses a mixture regularizer. The major differences with the work by Zimmert et al. [71] are that Algorithm 1 uses a log-barrier rather than Tsallis entropy, and that it takes advantages of optimistic predictors. As far as we have managed to determine, it appears difficult to combine Tsallis-entropy-based algorithms with optimistic predictors, as we discuss in Subsection 4.2. Our work overcomes this difficulty by using a log-barrier regularizer, rather than Tsallis entropy, in exchange for additional $O(\sqrt{\log T})$-factors in worst-case regret bounds.

**Remark 1.** In the previously mentioned study by Wei and Luo [68], the regret bounds for combinatorial semi-bandits are not given explicitly. However, as stated just after Corollary 3 in their paper ("but they can be straightforwardly generalized to the semi-bandit case"), we can obtain the regret bounds in Table 1 via a simple calculation. To be more precise, their regret bounds in Table 1 dependent on $Q_2$ and $V_1$ can be refined by replacing them with $Q_2(I^*) := \sum_{t=1}^{T} \sum_{i \in I^*} (\ell_{ti} - \bar{\ell}_i)^2$ and $V_2(I^*) := \sum_{t=1}^{T-1} \sum_{i \in I^*} |\ell_{ti} - \ell_{t+1,i}|$, respectively, where we define $I^* = \{i \in [d] \mid a_i^* = 1\}$ for an optimal action $a^*$. This means that Algorithm 1 is not necessarily superior to their algorithms. On the other hand, it should be noted that their algorithms require prior knowledge w.r.t. $Q_2(I^*)$ and $V_2(I^*)$ to achieve corresponding regret bounds.

For linear bandits problems, we provide the hybrid data-dependent regret bounds shown in Table 2 and in Theorem 3, which holds for $p \in [2, \infty]$ and $q \in [1, 2]$ such that $1/p + 1/q = 1$, under the assumption of $\|a\|_p \le 1$ for all $a \in \mathcal{A}$ and $\|\ell_t\|_q \le 1$ for all $t$. The parameter $\vartheta \ge 1$ is associated with a self-concordant barrier over the convex hull of $\mathcal{A}$. It is known that any convex set has a self-concordant barrier with $\vartheta = O(d)$ [53], which is tight up to a constant factor. Substituting $\vartheta = d$ and noting $L^* \le T$, we can see that the new regret bound includes previous bounds. Further, for the special case of $(p, q) = (\infty, 1)$, the new path-length regret bound of $\tilde{O}(d\sqrt{V_1})$ is tight up to a logarithmic factor in $T$, as a matching lower bound of $O(d\sqrt{V_1})$ is known [22, 15]. To our knowledge, this is the first (nearly) tight path-length regret bound for linear bandits. The regret bounds of $\tilde{O}(d\sqrt{L^*})$ and $\tilde{O}(d\sqrt{Q_q})$ are also nearly tight, as has been noted in the literature [36]. The approach for the new regret bound is quite simple: we combine regret bounds dependent on optimistic predictors [57, 36] and the algorithm of *tracking the best linear predictor* [31, 16].

## 2 Problem settings

This section introduces the problem settings of *combinatorial semi-bandits* and *linear bandits*. In both settings, a player is given, before the game starts, an *action set* $\mathcal{A} \in \mathbb{R}^d$ and the total number $T$ of rounds. In each round $t \in [T]$, the player chooses an action $a_t \in \mathcal{A}$, while the environment chooses a loss vector $\ell_t \in \mathbb{R}^d$. After choosing the action, the player gets feedback on the loss, which will depend on the problem settings. Player performance is measured in terms of regret $R_T$ defined as follows:

$$R_T(a^*) = \mathbf{E}\left[\sum_{t=1}^{T} \ell_t^\top (a_t - a^*)\right], \quad R_T = \max_{a^* \in \mathcal{A}} R_T(a^*), \tag{1}$$

where the expectation is taken w.r.t. the randomness of $\ell_t$ and the algorithm's internal randomness.

### 2.1 Combinatorial semi-bandits

This subsection provides settings of action sets and feedback information in combinatorial semi-bandits. The action set $\mathcal{A}$ is a subset of binary vectors $\{0, 1\}^d$, each element of which can be interpreted as a subset of $[d]$. Denote $m = \max_{a \in \mathcal{A}} \|a\|_1$. For each chosen action $a_t = [a_{t1}, a_{t2}, \ldots, a_{td}]^\top \in \mathcal{A}$, we denote $I_t = \{i \in [d] \mid a_{ti} = 1\}$. We further assume that $\ell_t \in [0, 1]^d$, similarly to what is seen in existing work [71, 40, 42, 66, 54].

In combinatorial semi-bandits, the player can get entry-wise bandit feedback. More precisely, after choosing an action $a_t$, which corresponds to a subset $I_t$ of $[d]$, the player can observe $\ell_{ti}$ for each $i \in I_t$, while $\ell_{ti}$ for $i \in J_t := [d] \setminus I_t$ will not be revealed.

In addition to general action set $\mathcal{A} \in \{0, 1\}^d$, this paper analyzes two special cases of settings. One is *size-invariant semi-bandits*, in which all actions $a \in \mathcal{A}$ have the same size $\|a\|_1 = m$. The other one is *matroid semi-bandits* [40, 66], in which an action set $\mathcal{A}$ corresponds to the bases of a matroid. As all bases of an arbitrary matroid have the same size, matroid semi-bandits are special cases of size-invariant semi-bandits, which implies

$$\{\text{general semi-bandits}\} \supseteq \{\text{size-invariant semi-bandits}\} \supseteq \{\text{matroid semi-bandits}\}.$$

An important example of matroid semi-bandits is the problem over $m$-set, which is defined as $\mathcal{A} = \{a \in \{0, 1\} \mid \|a\|_1 = m\}$. The problem with full-combinatorial set $\mathcal{A} = \{0, 1\}^n$ can also be reduced to a special case of matroid semi-bandits with $(d, m) = (2n, n)$. Zimmert et al. [71] have provided improved regret bounds for such special cases of $m$-sets and full-combinatorial sets.

## 2.2 Linear bandits

In linear bandits, the action set $\mathcal{A}$ is assumed to be an arbitrary closed and bounded subset of $\mathbb{R}^d$. The special cases in which $\mathcal{A}$ consists of binary vectors in $\{0,1\}^d$ are called *combinatorial bandits* [17]. Similarly to what has been done in existing work [15, 29], we assume that there exists $p, q \in [1, \infty]$ for which $1/p + 1/q = 1$, $\|a\|_p \leq 1$ and $\|\ell_t\|_q \leq 1$ hold for all $a \in \mathcal{A}$ and $\ell_t$. By rescaling $\mathcal{A}$ and $\{\ell_t\}$ as needed, any problem with bounded $\mathcal{A}$ and $\{\ell_t\}$ can be transformed into a problem satisfying this assumption.

The available feedback in linear bandits is even more limited than in combinatorial semi-bandits. After choosing an action $a_t \in \mathcal{A}$, the player can only observe the incurred loss $\ell_t^\top a_t$. In the special case of combinatorial bandits, the player can only observe the sum of losses $\sum_{i \in I_t} \ell_{ti}$ for the chosen subset, unlike in combinatorial semi-bandits in which $\ell_{ti}$ is revealed for each $i \in I_t$.

## 2.3 Assumptions regarding environments

The scope of this work includes the following three different settings in terms of the environments' determining losses $\ell_t$:

$$\{\text{adversarial regimes}\} \supseteq \{\text{stochastic regimes with adversarial corruptions}\} \supseteq \{\text{stochastic regiems}\}.$$

**Stochastic regimes**  In a stochastic regime, the loss vectors $\ell_t$ are supposed to follow an unknown distribution $\mathcal{D}$, i.i.d. for $t = 1, 2, \ldots, T$. Denote $\mu = \mathbf{E}_{\ell \sim \mathcal{D}}[\ell]$ and set $a^* \in \arg\min_{a \in \mathcal{A}} \mu^\top a$. The regret can then be expressed as $R_T = \mathbf{E}[\sum_{t=1}^T \mu^\top(a_t - a^*)]$. It is known that the optimal regret in this regime can be characterized by the *suboptimality gap* parameter $\Delta$ defined as $\Delta = \min_{a \in \mathcal{A} \setminus a^*} \mu^\top a - \mu^\top a^*$. Note that no prior information on the distribution $\mathcal{D}$, including the parameter $\Delta$, is given to the player. When we consider stochastic regime, we assume that $\Delta > 0$, which implies that the optimal action $a^* \in \arg\min_{a \in \mathcal{A}} \mu^\top a$ is assumed to be unique.

**Adversarial regimes**  In an adversarial regime, no stochastic models on $\ell_t$ are assumed, but the loss $\ell_t$ may be chosen in an adversarial manner. More precisely, the environment can choose $\ell_t$ depending on the actions and losses $\{(\ell_j, a_j)\}_{j=1}^{t-1}$ chosen up until the $(t-1)$-th round.

**Stochastic regimes with adversarial corruptions**  A stochastic regime with adversarial corruptions is a regime intermediate between stochastic regimes and adversarial regimes. In such a regime, a temporary loss $\ell_t'$ is drawn from an unknown distribution $\mathcal{D}$, and then the environment may corrupt it to determine $\ell_t$ in each round, subject to the constraint $\sum_{t=1}^T \|\ell_t - \ell_t'\|_\infty \leq C$, where $C \geq 0$ is a parameter called the *corruption level* and corresponds to the total amount of corruptions. In this paper, we suppose that the corruptions on $\ell_t$ depend on $\ell_t'$ and historical data $\{(\ell_j', \ell_j, a_j)\}_{j=1}^{t-1}$, and that they do not depend on $a_t$, similarly to what is seen in existing models [48, 27, 70, 8].

The special cases of the stochastic regime with adversarial corruptions in which $C \geq \Omega(T)$ and $C = 0$ coincide, respectively, with the adversarial regime and the stochastic regime. This paper supposes that the player is not given parameter $C$ in advance, i.e., it aims to adapt to any environment with an arbitrary corruption level $C$.

# 3 Related work

Combinatorial semi-bandits have been extensively studied for a wide range of applications, including adaptive routing [25], network optimization [40], spectrum allocations [25] and recommender systems [41, 65, 56]. For stochastic combinatorial semi-bandits, Kveton et al. [42, 40], Wang and Chen [66] provide tight regret bounds dependent on the suboptimality gap. Interestingly, these tight regret bounds differ depending on the assumption of the action set: for the general action set, the tight bound is of $O(\frac{dm \log T}{\Delta})$, while, in the matroid semi-bandit cases, the tight bound is of $O(\frac{(d-m) \log T}{\Delta})$. Similar tight bounds are reproduced in the work by [71] and in this work as well, together with worst-case optimal regret bounds for the adversarial setting. Chen et al. [19, 20] have considered a more extended framework including nonlinear reward functions. Linear bandits also have many applications, including end-to-end adaptive routing [7, 6] and various examples of combinatorial bandits [17].

In a bandits context, BOBW algorithms have been developed for various problem settings, including the multi-armed bandit [12, 62, 5, 61, 68, 70, 59], combinatorial semi-bandits [71], episodic Markov decision processes [37, 38], online learning with feedback graphs [24], and linear bandits [45]. Similar algorithms have been developed for full-information online learning problems as well, such as the problem of prediction with expert advice [2, 52, 23, 26] and online linear optimization [32]. For achieving BOBW, two main approaches can be found in these papers. One is to select an appropriate mode in an online manner, by determining whether the environment is i.i.d. or not. The other is to exploit self-bounding constraints, i.e., an approach which is meant to lead to improved bounds by combining a regret *lower* bound expressed with a suboptimality gap and a regret upper bound dependent on the action probability vectors. This work adopts the latter approach, similarly to certain existing work [70, 71, 68].

Since Lykouris et al. [48] initiated a study on stochastic bandits robust to adversarial corruptions, research in this direction has been extended to a variety of models, such as those for (adversarial) multi-armed bandits [27, 70, 28, 50], episodic Markov decision processes [49, 21, 38], Gaussian process bandits [8], the problem of prediction with expert advice [2, 33], online learning with feedback graphs [24], and linear bandits [9, 45]. There can be found studies on effective attacks and on the vulnerability of well-known algorithms [39, 46]. We note that some existing studies (e.g., [9, 39, 28, 46]) have considered *targeted corruption* models, in which the adversary may choose corruption on $\ell_t$ after observing the player's action $a_t$, unlike this work and some previous studies [48, 27, 70, 8]. The differences in corruption models can be summarized; see, e.g., the paper by Hajiesmaili et al. [28]. In addition, there are alternative definitions of regret, e.g., as one is defined with the losses *after* corruptions, as in this work and certain previous studies [48, 70, 8] and another is defined with the losses *without* corruptions [27, 45, 9]. As the gap between these two notions of regret is at most $O(C)$, regret bounds for one side has consequences for the other sides up to an additional $O(C)$-term. For further discussion on alternative notions of regret, see, e.g., Section 5.2 of the paper by Gupta et al. [27].

Data-dependent bounds have been studied for a variety of difficulty indicators. For a bandits context, Allenberg et al. [1] have developed a multi-armed bandit algorithm with a first-order regret bound, i.e., the bound dependent on $L^*$ rather than on $T$. Hazan and Kale [29] provided algorithms so-called second-order regret bounds, which depend on $Q_2$. Similarly to what is seen in such full-information online learning problems as the problem of prediction with expert advice, there are algorithms with first- and second-order regret bounds [18, 26, 63, 47]. It is worth mentioning that one kind of second-order regret bound implies BOBW guarantees, as shown by Gaillard et al. [26]. Note that some known difficulty indicators are not dealt with in this work, e.g., the sparsity of loss vectors [43, 14].

# 4 Combinatorial semi-bandits

## 4.1 Preliminary: existing techniques

**Convex combination and decomposition**  Let $\mathcal{X}$ denote the convex hull of the action set $\mathcal{A}$, i.e., the set of all vectors that can be expressed by a convex combination of vectors in $\mathcal{A}$. Our proposed algorithm manages vectors $x_t \in \mathcal{X}$, and chooses $a_t \in \mathcal{A}$ so that $\mathbf{E}[a_t|x_t] = x_t$. Such $a_t$ can be generated via a convex decomposition of $x_t$. In fact, from Carathéodory's theorem, for any $x_t \in \mathcal{X}$, there exist $\{\lambda_k\}_{k=0}^d \subseteq [0,1]$ and $\{a_t^{(k)}\}_{k=0}^d \subseteq \mathcal{A}$ such that $\sum_{k=0}^d \lambda_k = 1$ and $\sum_{k=0}^d \lambda_k a_t^{(k)} = x_t$. Hence, by choosing $a_t = a_t^{(k)}$ with probability $\lambda_k$, we have $\mathbf{E}[a_t|x_t] = x_t$. Such $\{\lambda_k\}_{k=0}^d$ and $\{a_t^{(k)}\}_{k=0}^d$ can be computed efficiently if there is an algorithm for solving linear optimization over $\mathcal{A}$, as shown, e.g., in Corollary 11.4 in [60]. Similar techniques are used in [35, 36] as well. More efficient algorithms for computing $\{\lambda_k\}_{k=0}^d$ have been developed for the special cases of bases of uniform matroids [71, 67] and general matroids [64]. In our regret analyses, we use the fact that $I_t = \{i \in [d] \mid a_{ti} = 1\}$ satisfies $\mathrm{Prob}[i \in I_t|x_t] = x_{ti}$.

**Optimistic follow the regularized leader**  Our proposed algorithm is based on the framework of *optimistic follow the regularized leader* [57, 58, 44], in which the vector $x_t$ is defined as

$$x_t \in \underset{x \in \mathcal{X}}{\arg\min} \left\{ \sum_{j=1}^{t-1} \hat{\ell}_j^\top x + m_t^\top x + \psi_t(x) \right\}, \tag{2}$$

where each $\hat{\ell}_j$ is an unbiased estimator of $\ell_j$, $m_t$ is an arbitrary *hint vector* estimating $\ell_t$, and $\psi_t$ is a regularizer that is a smooth convex function over $\mathcal{X}$. The regret for this algorithm can be evaluated by means of Bregman divergences defined by $D_t(x, y) = \psi_t(x) - \psi_t(y) + \nabla \psi_t(y)^\top (x - y)$.

**Lemma 1.** *If $x_t$ is given by* (2)*, we then have* $\sum_{t=1}^T \hat{\ell}_t^\top (x_t - x^*) \leq \psi_{T+1}(x^*) - \psi_1(x_1') + \sum_{t=1}^T \left( (\hat{\ell}_t - m_t)^\top (x_t - x_{t+1}') - D_t(x_{t+1}', x_t) + \psi_t(x_{t+1}') - \psi_{t+1}(x_{t+1}') \right)$*, where $x_t'$ is defined as* $x_t' \in \arg\min_{x \in \mathcal{X}} \left\{ \sum_{j=1}^{t-1} \hat{\ell}_j^\top x + \psi_t(x) \right\}$*.*

All omitted proofs are given in the Appendix. A similar framework is used in [36] for linear bandits. Further, a special case in which $m_t = 0$ has been employed in [71].

**Remark 2.** In some existing work, a slightly different approach called online mirror descent has been used, e.g., in [68]. In online mirror descent, the update rule is expressed as $x_t \in \arg\min_{x \in \mathcal{X}} \left\{ \hat{\ell}_t^\top x + D_t(x, x_{t-1}) \right\}$. The relationship between follow the regularized leader and online mirror descent has been widely discussed [51, 44]. Amir et al. [2] have pointed out an essential difference: an algorithm in the follow-the-regularized-leader framework has improved performance in stochastic regimes, but none in online mirror descent has done so.

## 4.2 Proposed algorithm

In our proposed algorithm, we define an unbiased estimator $\hat{\ell}_t$ and regularizer $\psi_t$ as follows:

$$\hat{\ell}_{ti} = m_{ti} + \frac{a_{ti}}{x_{ti}}(\ell_{ti} - m_{ti}), \quad \psi_t(x) = \sum_{i=1}^d \beta_{ti} \left( -\log x_i + \gamma(1 - x_i)\log(1 - x_i) \right), \tag{3}$$

where $\beta_{ti}$ and $\gamma$ are defined as

$$\alpha_{ti} = a_{ti}(\ell_{ti} - m_{ti})^2 \min\left\{ 1, \frac{1 - x_{ti}}{\gamma x_{ti}^2} \right\}, \quad \beta_{ti} = \sqrt{2 + \frac{1}{\log T} \sum_{j=1}^{t-1} \alpha_{ji}}, \quad \gamma = \log T. \tag{4}$$

Our hybrid regularizer given in (3) is designed to lead to improved regret bounds in stochastic settings. In order to show BOTW regret bounds, it is necessary that round-wise regret bounds (e.g., the stability term in the paper by Zimmert et al. [71]) converge to 0 when $x_t$ approaches extreme points in $\{0, 1\}^d$. As can be seen in Lemma 2 below, the round-wise regret bounds of our algorithm can be expressed as $O(\sum_{i=1}^d \frac{\alpha_{ti}}{\beta_{ti}}) = O(\sum_{i=1}^d \frac{x_{ti}}{\beta_{ti}} \min\{1, \frac{1-x_{ti}}{\gamma x_{ti}^2}\})$ in expectation. Without hybrid regularization, i.e., if $\gamma = 0$, we cannot obtain the above-mentioned convergence property, particularly when $x_{ti}$ approaches 1 for some $i$'s. On the other hand, thanks to hybrid regularization (with $\gamma > 0$), we can show that the round-wise regret converge to 0 when approaching any points in $\{0, 1\}^d$. The learning rate parameters $\beta_{ti}$ given in (4) are designed so that two main parts of the regret bound, $\sum_{t=1}^T \frac{\alpha_{ti}}{\beta_{ti}}$ and $\log T \cdot \beta_{T+1,i}$, will be well-balanced.

The optimistic predictor $m_t$ is updated as follows:

$$m_{1i} = 1/4 \quad (i \in [d]), \quad m_{t+1,i} = m_{ti} + a_{ti}(\ell_{ti} - m_{ti})/4 \quad (t \in [T], i \in [d]). \tag{5}$$

The proposed algorithm can be summarized in Algorithm 1, which is similar to the one proposed in [71] in that both are based on the follow-the-regularized-leader framework with a round-dependent regularizer. The main differences are as follows:

- Algorithm 1 employs an optimistic-prediction framework while the algorithm in [71] does not, i.e., $m_t$ is fixed to the zero vector for each $t$.

---

**Algorithm 1** Hybrid algorithm for combinatorial semi-bandits

---

**Require:** Action set $\mathcal{A}$, time horizon $T \in \mathbb{N}$
1: Initialize $m_t \in [0,1]^d$ by $m_{1i} = 1/2$ for all $i \in [d]$.
2: **for** $t = 1, 2, \ldots, T$ **do**
3:     Compute $x_t$ as (2), where $\hat{\ell}_j$, $\psi_t$ and $\beta_{ti}$ are defined in (3) and (4), respectively.
4:     Pick $a_t$ so that $\mathbf{E}[a_t|x_t] = x_t$, output $a_t$, and get feedback of $\ell_{ti}$ for each $i$ such that $a_{ti} = 1$.
5:     Compute $\hat{\ell}_t$, $\beta_{t+1,i}$ and $m_{t+1}$ on the basis of (3), (4) and (5), respectively.
6: **end for**

---

- Algorithm 1 uses a hybrid regularizer combining the log-barrier and Shannon entropy given in (3), while Zimmert et al. [71] adopt the combination of Tsallis entropy with power $1/2$ and Shannon entropy defined as $\psi_t(x) = \frac{1}{\sqrt{t}} \sum_{i=1}^d (-\sqrt{x_i} + \gamma(1-x_i)\log(1-x_i))$.

- Algorithm 1 maintains the strength $\gamma_{ti}$ for regularization that is different for each entry, and it updates each on the basis of historical data $\{x_j, a_j, (\ell_{ji})_{i \in I_t}\}_{j=1}^{t-1}$.

The reason for using a log-barrier regularizer is that it allows us to exploit the optimistic prediction framework, i.e., it provides regret bounds dependent on $(\ell_t - m_t)$. When $m_t$ are non-zero vectors, $\hat{\ell}_t$ can be $O(1/x_{ti})$ negative values, which would make it even more difficult to bound the regret if Tsallis entropy were used. For details, see, e.g., [69] and the paragraph just after (RV) in [70]. In contrast to this, a log-barrier regularizer works well even for $O(1/x_{ti})$ negative losses, which is convenient for combining with an optimistic prediction framework.

### 4.3 Regret analysis

This subsection provides regret bounds achieved by Algorithm 1. First, as we have $\mathbf{E}[\hat{\ell}|x_t] = \ell_t$ from (3) and $\mathbf{E}[a_t|x_t] = x_t$, the regret can be bounded as

$$R_T(a^*) \leq \mathbf{E}\left[\sum_{t=1}^T \ell_t^\top (x_t - x^*)\right] + T\|a^* - x^*\|_1 = \mathbf{E}\left[\sum_{t=1}^T \hat{\ell}_t^\top (x_t - x^*)\right] + T\|a^* - x^*\|_1 \quad (6)$$

for any $x^* \in \mathcal{X}$. We set $x^* = (1 - \frac{d}{T})a^* + \frac{d}{T}x_0$, where $x_0$ is a point in $\mathcal{X}$ such that $x_{0i} \geq 1/d$ for all $i \in [d]$. The existence of such a point follows from the assumption that for any $i \in [d]$ there exists $a \in \mathcal{A}$ satisfying $a_i = 1$. The term $\|a^* - x^*\|_1$ in (6) can then be bounded as $\|a^* - x^*\|_1 = \frac{d}{T}\|a^* - x_0\|_1 \leq \frac{d^2}{T}$. Further, the term $\sum_{t=1}^T \hat{\ell}_t^\top (x_t - x^*)$ can be bounded via Lemma 1 and the following lemma:

**Lemma 2.** *Suppose $\hat{\ell}_t$ and $\psi_t$ are given by (3), respectively. The following part of the bound in Lemma 1 can then be bounded as $(\hat{\ell}_t - m_t)^\top (x_t - x'_{t+1}) - D_t(x'_{t+1}, x_t) = O\left(\sum_{i=1}^d \frac{\alpha_{ti}}{\beta_{ti}}\right)$, where $\alpha_{ti}$ is defined in (4).*

This lemma can be shown via standard techniques used, e.g., in [71, 68]. Combining Lemmas 1, 2 and (6), we obtain the following regret bound:

**Theorem 1.** *For Algorithm 1, the regret is bounded as $R_T = O\left(\log T \cdot \mathbf{E}\left[\sum_{i=1}^d \beta_{T+1,i}\right] + d^2\right)$ where $\beta_{ti}$ is defined as (4). Consequently, we have*

$$R_T = O\left(\sum_{i=1}^d \sqrt{\log T \, \mathbf{E}\left[\sum_{t=1}^T a_{ti}(\ell_{ti} - m_{ti})^2\right]} + d^2 + d\log T\right) \quad (7)$$

*as well as*

$$R_T = O\left(\sum_{i=1}^d \sqrt{\log T \, \mathbf{E}\left[\sum_{t=1}^T \min\left\{x_{ti}, \frac{1-x_{ti}}{\sqrt{\log T}}\right\}\right]} + d^2 + d\log T\right). \quad (8)$$

Note that this theorem holds for arbitrary $m_t \in [0,1]^d$. The specific choice of $m_t$ in (5) and (7) leads to the following bound in the adversarial regime:

**Corollary 1** (Data-dependent bounds for adversarial regimes)**.** *For Algorithm 1, the regret is bounded as* $R_T = O\left(\sqrt{d \log T \cdot \min\left\{\sum_{t=1}^{T} \ell_t^\top a^*, \sum_{t=1}^{T} \|\ell_t - \bar{\ell}\|_2^2, \sum_{t=1}^{T-1} \|\ell_t - \ell_{t+1}\|_1\right\}} + d \log T + d^2\right)$
*for any* $a^* \in \mathcal{A}$*, where* $\bar{\ell} = \frac{1}{T} \sum_{t=1}^{T} \ell_t$*.*

Further, from (8), we see that Algorithm 1 may offer improved performance in stochastic regimes (with adversarial corruptions), as follows:

**Corollary 2** (Improved regret bounds for stochastic regimes with adversarial crruptions)**.** *If the environment is in a stochastic regime with adversarial corruptions (defined in Subsection 2.3), Algorithm 1 has the following regret bound:* $R_T(a^*) = O\left(\frac{B(\mathcal{A}) \log T}{\Delta} + \sqrt{B(\mathcal{A}) \frac{Cm \log T}{\Delta}} + d^2\right)$*,*
*where* $B(\mathcal{A}) \geq 0$ *is a constant dependent on the action set, bounded as*

$$B(\mathcal{A}) \leq \begin{cases} dm & (\textit{general cases}) \\ (d - m + m/\sqrt{\log T}) \min\{m, d - m\} & (\textit{size-invariant semi-bandits}) \\ d - m + m/\sqrt{\log T} & (\textit{matroid semi-bandits}) \end{cases} \quad (9)$$

**Remark 3.** From Corollary 2, we can obtain regret bounds for the stochastic regime as well, by substituting $C = 0$. In a stochastic regime, the BOBW algorithm proposed by Zimmert et al. [71] has been shown to enjoy similar but slightly different regret bounds, e.g., $B(\mathcal{A}) \leq (d + m/\log T)m$ for general cases (which is slightly worse than in (9)), and $B(\mathcal{A}) \leq (d - m)(1 + (\log d)^2/\log T)$ for the cases of uniform matroids (which in general is not comparable to (9)). For a stochastic regime with adversarial corruptions, their algorithm achieves $O(\frac{B(\mathcal{A}) \log T}{\Delta} + Cm)$-regret for such a modified $B(\mathcal{A})$, though it is not known if the bound can be improved to an $O(\sqrt{C})$-type as in Corollary 2.

**Remark 4.** For matroid semi-bandits, we can state a more refined regret bound. For the optimal action $a^* \in \arg\min_{a \in \mathcal{A}} \mu^\top a$ set $J^* = \{i \in [d] \mid a_i^* = 0\}$ and denote $\Delta = \min_{a \in \mathcal{A}: a_i = 1} \mu^\top a - \mu^\top a^*$ for each $i \in J^*$. We then have $R_T(a^*) = O\left(\sum_{i \in J^*} \frac{\log T}{\Delta_i} + \frac{m\sqrt{\log T}}{\Delta} + \sqrt{Cm\left(\sum_{i \in J^*} \frac{\log T}{\Delta_i} + \frac{m\sqrt{\log T}}{\Delta}\right)}\right)$.

# 5 Linear bandits

## 5.1 Predictor-dependent regret bounds

Regret bounds dependent on $m_t$ have been developed for linear bandit problems, similarly to what is seen for semi-bandits. This paper focuses on regret bounds in the following form:

$$R_T \leq D \cdot \mathbf{E}\left[\sqrt{\sum_{t=1}^{T} ((\ell_t - m_t)^\top a_t)^2}\right], \quad (10)$$

where $D$ is a parameter dependent on $\mathcal{A}, d$ and $T$. Rakhlin and Sridharan [57] proposed the SCRiBLe algorithm, which achieves a regret bound as in (10) with $D = O(\vartheta d \log T)$, given a $\vartheta$-self-concordant barrier over the convex hull of $\mathcal{A}$, if an appropriate learning rate is chosen. Even without self-concordant barriers, for general action sets, an algorithm proposed by Ito et al. [36] achieves a regret bound with $D = O(d \log T \cdot \log(dT))$, as is shown in Theorem 2 in their paper.

From (10), we can achieve small regret by choosing $m_t$ so that $\sum_{t=1}^{T} g_t(m_t)$, where we define $g_t(m) = \frac{1}{2}((\ell_t - m)^\top a_t)^2$. When choosing $m_t$, we can use the information of $\{(\ell_j, \ell_j^\top a_j)\}_{j=1}^{t-1}$.

## 5.2 Tracking linear experts

In this subsection, we revisit the work by Herbster and Warmuth [31]. Let $p' = \min\{p, 2 \log d\}$ and let $q' \geq q$ be such that $1/p' + 1/q' = 1$. Define $\Phi_p(m) = \|m\|_p^2$. Let $\mathcal{L} = \{m \in \mathbb{R}^d \mid \|m\|_q \leq 1\}$. Consider $m_t \in \mathcal{L}$ defined as follows:

$$m_1 = 0, \quad m_{t+1} \in \arg\min_{m \in \mathcal{L}} \left\{\Phi_{p'}(m) + \left(\nabla \Phi_{q'}(m_t) - \frac{1}{2(p' - 1)}\left(m_t^\top a_t - \ell_t^\top a_t\right) a_t\right)^\top m\right\}. \quad (11)$$

For $m_t$ defined by (11), we have the following:

**Theorem 2** ([31], Theorem 11.4 in [16]). *Suppose $p \geq 2$. If $m_t$ is chosen by (11), for any sequence $\{u_t\}_{t=1}^T \subseteq \mathcal{L}$, we have $\sum_{t=1}^T g_t(m_t) \leq 2 \sum_{t=1}^T g_t(u_t) + 4(p'-1)\left(\sum_{t=1}^T \|u_t - u_{t+1}\|_q + 1\right).$*

### 5.3 Hybrid data-dependent regret bound

By combining (10) and Theorem 2, we obtain the following regret bound:

**Theorem 3.** *Suppose $p \geq 2$. Suppose an algorithm enjoys a regret bound as (10) and $m_t$ is chosen by (11). We then have $R_T = O\left(D \cdot \mathbf{E}\left[\sqrt{\min\{Q_q, p' \cdot V_q\} + p'}\right]\right)$ for any $u \in \mathcal{L}$. Further, if $\ell_t^\top a \geq 0$ holds for all $a \in \mathcal{A}$ and all $t \in [T]$, it holds for all $a^* \in \mathcal{A}$ that $R_T(a^*) = O\left(D \cdot \sqrt{\mathbf{E}[L^*] + p'}\right).$*[2]

By combining this theorem and (10) with $D = \tilde{O}(d)$ [36], we obtain the regret bound in Table 2.

**On Potential Societal Impact**   This study is primarily theoretical in nature, and we do not see any negative social consequences. Researchers working on bandit theory may benefit from this paper. In the long run, we expect that the proposed algorithms, which are robust to adversarial attacks, have the potential to contribute to the realization of a safer and more secure society.

## Acknowledgments and Disclosure of Funding

The author was supported by JST, ACT-I, Grant Number JPMJPR18U5, Japan.

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
