# A Appendix

## A.1 Proof of Lemma 1

From the definition of $x_t$ and $x'_t$ shown in (2) and the first-order optimality condition, we have

$$\sum_{t=1}^{T} \hat{\ell}_t^\top x^* + \psi_{T+1}(x^*) \geq \sum_{t=1}^{T} \hat{\ell}_t^\top x'_{T+1} + \psi_{T+1}(x'_{T+1})$$

$$= \left(\sum_{t=1}^{T-1} \hat{\ell}_t + m_T\right)^\top x'_{T+1} + \psi_T(x'_{T+1}) + \psi_{T+1}(x'_{T+1}) - \psi_T(x'_{T+1}) + (\hat{\ell}_T - m_T)^\top x'_{T+1}$$

$$\geq \left(\sum_{t=1}^{T-1} \hat{\ell}_t + m_T\right)^\top x_T + \psi_T(x_T) + \psi_{T+1}(x'_{T+1}) - \psi_T(x'_{T+1}) + (\hat{\ell}_T - m_T)^\top x'_{T+1} + D_T(x'_{T+1}, x_T)$$

$$= \sum_{t=1}^{T-1} \hat{\ell}_t^\top x_T + \psi_T(x_T) + m_T^\top x_T + \psi_{T+1}(x'_{T+1}) - \psi_T(x'_{T+1}) + (\hat{\ell}_T - m_T)^\top x'_{T+1} + D_T(x'_{T+1}, x_T)$$

$$\geq \sum_{t=1}^{T-1} \hat{\ell}_t^\top x'_T + \psi_T(x'_T) + m_T^\top x_T + \psi_{T+1}(x'_{T+1}) - \psi_T(x'_{T+1}) + (\hat{\ell}_T - m_T)^\top x'_{T+1} + D_T(x'_{T+1}, x_T)$$

$$\geq \cdots \geq \psi_1(x'_1) + \sum_{t=1}^{T} \left(\psi_{t+1}(x'_{t+1}) - \psi_t(x'_{t+1}) + m_t^\top x_t + (\hat{\ell}_t - m_t)^\top x'_{t+1} + D_t(x'_{t+1}, x_t)\right),$$

where the first and the third inequalities follow from the definition of $x'_t$, the second inequality follows from the definition (2) of $x_t$ and the first-order optimality condition, and the last inequality is obtained by applying similar arguments recursively. This inequality immediately implies the bound in Lemma 1. $\square$

For the special case in which $m_t = 0$, a proof can be found in the literature, e.g., in Exercise 28.12 of the book by Lattimore and Szepesvári [44].

## A.2 Proof of Lemma 2

For the sake of simplicity, we here assume $T \geq 3$ and, consequently, have $\gamma = \log T \geq 1$. For each $i \in [d]$, we denote

$$\psi_{ti}^{(1)}(x_{ti}) = -\beta_{ti} \log(x_i), \quad \psi_{ti}^{(2)}(x) = \gamma \beta_{ti}(1 - x_i) \log(1 - x_i) \tag{12}$$

and let $D_{ti}^{(1)}$ and $D_{ti}^{(2)}$ be the Bregman divergences over $\mathbb{R}_{>0}$, corresponding to $\psi_{ti}^{(1)}$ and $\psi_{ti}^{(2)}$, respectively. As $\psi_t$ can be expressed as $\psi_t(x) = \sum_{i=1}^{d}(\psi_{ti}^{(1)}(x_i) + \psi_{ti}^{(2)}(x_i))$, from the linearity of Bregman divergences, we have $D_t(x, y) = \sum_{i=1}^{d}(D_{ti}^{(1)}(x_i, y_i) + D_{ti}^{(2)}(x_i, y_i))$. We hence have

$$(\hat{\ell}_t - m_t)^\top (x_t - x'_{t+1}) - D_t(x'_{t+1}, x_t)$$

$$= \sum_{i=1}^{d} \left((\hat{\ell}_{ti} - m_{ti})(x_{ti} - x'_{t+1,i}) - D_{ti}^{(1)}(x'_{t+1,i}, x_{ti}) - D_{ti}^{(1)}(x'_{t+1,i}, x_{ti})\right)$$

$$\leq \sum_{i=1}^{d} \min \left\{(\hat{\ell}_{ti} - m_{ti})(x_{ti} - x'_{t+1,i}) - D_{ti}^{(1)}(x'_{t+1,i}, x_{ti}), (\hat{\ell}_{ti} - m_{ti})(x_{ti} - x'_{t+1,i}) - D_{ti}^{(2)}(x'_{t+1,i}, x_{ti})\right\}. \tag{13}$$

We show below that

$$(\hat{\ell}_{ti} - m_{ti})^\top (x_{ti} - x) - D_{ti}^{(1)}(x, x_{ti}) \leq \beta_{ti} h^{(1)}\left(\frac{a_{ti}(\ell_{ti} - m_{ti})}{\beta_{ti}}\right) \tag{14}$$

$$(\hat{\ell}_{ti} - m_{ti})^\top (x_{ti} - x) - D_{ti}^{(2)}(x, x_{ti}) \leq \gamma \beta_{ti}(1 - x_{ti}) h^{(2)}\left(\frac{a_{ti}(\ell_{ti} - m_{ti})}{\gamma \beta_{ti} x_{ti}}\right) \tag{15}$$

hold for any $x \in \mathbb{R}_{>0}$, where we define $h^{(1)}(z) = z - \log(z+1)$ and $h^{(2)}(z) = \exp(z) - z - 1$.

Let us first show (14). From the first-order optimality condition, the left-hand side of (14) is maximized when $x$ satisfies

$$-\hat{\ell}_{ti} + m_{ti} - \nabla\psi_{ti}^{(1)}(x) + \nabla\psi_{ti}^{(1)}(x_{ti}) = 0, \tag{16}$$

which can be rewritten as

$$\frac{1}{x} = \frac{1}{x_{ti}} + \frac{\hat{\ell}_{ti} - m_{ti}}{\beta_{ti}}, \quad \text{and equivalently,} \quad \frac{x_{ti}}{x} = 1 + \frac{x_{ti}(\hat{\ell}_{ti} - m_{ti})}{\beta_{ti}}. \tag{17}$$

For such $x$, the left-hand side of (14) can be expressed as

$$(\hat{\ell}_{ti} - m_{ti})(x_{ti} - x) - D_{ti}^{(1)}(x, x_{ti})$$

$$= (\hat{\ell}_{ti} - m_{ti})(x_{ti} - x) + \beta_{ti}\left(\log x - \log x_{ti} - \frac{1}{x_{ti}}(x - x_{ti})\right)$$

$$= \left(\hat{\ell}_{ti} - m_{ti} + \frac{\beta_{ti}}{x_{ti}}\right)(x_{ti} - x) + \beta_{ti}\left(-\log\left(\frac{x_{ti}(\hat{\ell}_{ti} - m_t)}{\beta_{ti}} + 1\right)\right)$$

$$= \frac{\beta_{ti}}{x}(x_{ti} - x) + \beta_{ti}\left(-\log\left(\frac{x_{ti}(\hat{\ell}_{ti} - m_{ti})}{\beta_{ti}} + 1\right)\right)$$

$$= \beta_{ti}\left(-\log\left(\frac{x_{ti}(\hat{\ell}_{ti} - m_{ti})}{\beta_{ti}} + 1\right) + \frac{x_{ti}(\hat{\ell}_{ti} - m_{ti})}{\beta_{ti}}\right)$$

$$= \beta_{ti}h^{(1)}\left(\frac{x_{ti}(\hat{\ell}_{ti} - m_{ti})}{\beta_{ti}}\right) = \beta_{ti}h^{(1)}\left(\frac{a_{ti}(\ell_{ti} - m_{ti})}{\beta_{ti}}\right),$$

where the second, third, and fourth equalities follow from (17), and the last equality follows from the definition of $\hat{\ell}_t$ in (3).

Let us next show (15). From the first-order optimality condition, the left-hand side of (15) is maximized when $x$ satisfies

$$\log(1 - x) = \log(1 - x_{ti}) + \frac{\hat{\ell}_{ti} - m_{ti}}{\gamma\beta_{ti}}. \tag{18}$$

The left-hand side of (15) can then be expressed as

$$(\hat{\ell}_{ti} - m_{ti})(x_{ti} - x) - D_{ti}^{(2)}(x, x_{ti})$$
$$= (\hat{\ell}_{ti} - m_{ti} + \gamma\beta_{ti}(\log(1 - x_{ti}) + 1))(x_{ti} - x_i) - \gamma\beta_{ti}((1 - x_i)\log(1 - x_i) - (1 - x_{ti})\log(1 - x_{ti}))$$
$$= \gamma\beta_{ti}((\log(1 - x_i) + 1) \cdot (x_{ti} - x_i) - ((1 - x_i)\log(1 - x_i) - (1 - x_{ti})\log(1 - x_{ti})))$$
$$= \gamma\beta_{ti}((x_{ti} - 1)\log(1 - x_i) + (1 - x_{ti})\log(1 - x_{ti}) + x_{ti} - x_i)$$
$$= \gamma\beta_{ti}(1 - x_{ti})\left(\log\frac{1 - x_{ti}}{1 - x_i} - 1 + \frac{1 - x_i}{1 - x_{ti}}\right)$$
$$= \gamma\beta_{ti}(1 - x_{ti})\left(\exp\left(\frac{\hat{\ell}_{ti} - m_{ti}}{\gamma\beta_{ti}}\right) - \frac{\hat{\ell}_{ti} - m_{ti}}{\gamma\beta_{ti}} - 1\right)$$
$$= \gamma\beta_{ti}(1 - x_{ti})h^{(2)}\left(\frac{\hat{\ell}_{ti} - m_{ti}}{\gamma\beta_{ti}}\right) = \gamma\beta_{ti}(1 - x_{ti})h^{(2)}\left(\frac{a_{ti}(\ell_{ti} - m_{ti})}{\gamma\beta_{ti}x_{ti}}\right).$$

Combining (13), (14) and (15), we obtain

$$(\hat{\ell}_t - m_t)^\top(x_t - x'_{t+1}) - D_t(x'_{t+1}, x_t)$$

$$\leq \sum_{i=1}^{d} \beta_{ti} \min\left\{h^{(1)}\left(\frac{a_{ti}(\ell_{ti} - m_{ti})}{\beta_{ti}}\right), \gamma(1 - x_{ti})h^{(2)}\left(\frac{a_{ti}(\ell_{ti} - m_{ti})}{\gamma\beta_{ti}x_{ti}}\right)\right\}. \tag{19}$$

As we have $h^{(1)}(z) \leq 2z^2$ for $|z| \leq \frac{1}{\sqrt{2}}$ and $h^{(2)}(z) \leq z^2$ for $|z| \leq \sqrt{2}$, from $\beta_{ti} \geq \sqrt{2}$, we have

$$\min\left\{h^{(1)}\left(\frac{a_{ti}(\ell_{ti} - m_{ti})}{\beta_{ti}}\right), \gamma(1 - x_{ti})h^{(2)}\left(\frac{a_{ti}(\ell_{ti} - m_{ti})}{\gamma\beta_{ti}x_{ti}}\right)\right\}$$

$$\leq \begin{cases} \frac{2a_{ti}(\ell_{ti} - m_{ti})^2}{\beta_{ti}^2} & (\gamma\beta_{ti}x_{ti} < \frac{1}{\sqrt{2}}) \\ \min\left\{\frac{2a_{ti}(\ell_{ti} - m_{ti})^2}{\beta_{ti}^2}, \frac{a_{ti}(1 - x_{ti})(\ell_{ti} - m_{ti})^2}{\gamma\beta_{ti}^2 x_{ti}^2}\right\} & (\gamma\beta_{ti}x_{ti} \geq \frac{1}{\sqrt{2}}) \end{cases}$$

$$\leq \frac{2a_{ti}(\ell_{ti} - m_{ti})^2}{\beta_{ti}^2}\min\left\{1, \frac{1 - x_{ti}}{\gamma x_{ti}^2}\right\} = \frac{2\alpha_{ti}}{\beta_{ti}^2}.$$

The last inequality can be confirmed as follows: If $\frac{1 - x_{ti}}{\gamma x_{ti}^2} \leq 1$, then $1 - x_{ti} \leq \gamma x_{ti}^2 \leq \gamma x_{ti}$, which implies $x_{ti} \geq \frac{1}{\gamma+1} \geq \frac{1}{2\gamma} \geq \frac{1}{\sqrt{2}\gamma\beta_{ti}}$. Combining this with (19), we obtain

$$(\hat{\ell}_t - m_t)^\top(x_t - x'_{t+1}) - D_t(x'_{t+1}, x_t) \leq 2\sum_{i=1}^{d}\frac{\alpha_{ti}}{\beta_{ti}}.$$

$\square$

## A.3 Proof of Theorem 1

From (6), the definition of $x^*$, and Lemmas 1 and 2, we have

$$R_T(a^*) \leq \mathbf{E}\left[\psi_{T+1}(x^*) - \psi_1(x'_1) + 2\sum_{t=1}^{T}\left(\psi_t(x'_{t+1}) - \psi_{t+1}(x'_{t+1}) + \sum_{i=1}^{d}\frac{\alpha_{ti}}{\beta_{ti}}\right)\right] + d^2$$

$$\leq \mathbf{E}\left[\sum_{i=1}^{d}\beta_{T+1,i}\log\frac{1}{x_i^*} + \frac{\gamma}{e}\sum_{i=1}^{d}\beta_{1i} + 2\sum_{t=1}^{T}\sum_{i=1}^{d}\left(\frac{\alpha_{ti}}{\beta_{ti}} + \frac{\gamma}{e}(\beta_{t+1,i} - \beta_{ti})\right)\right] + d^2$$

$$\leq \mathbf{E}\left[\left(\log T + \frac{\gamma}{e}\right)\sum_{i=1}^{d}\beta_{T+1,i} + 2\sum_{i=1}^{d}\sum_{t=1}^{T}\frac{\alpha_{ti}}{\beta_{ti}}\right] + d^2, \tag{20}$$

where the second inequality follows from $\log x'_{ti} - \gamma(1 - x'_{ti})\log(1 - x'_{ti}) \leq \gamma/e$ as $x'_{ti} \in (0, 1)$ and the last inequality follows from the definition of $x^*$. We further have

$$\sum_{t=1}^{T}\frac{\alpha_{ti}}{\beta_{ti}} \leq 2\log T \cdot \beta_{T+1,i}. \tag{21}$$

In fact, for $\beta'_{ti}$ defined by $\beta'_{ti} = \sqrt{\frac{1}{\log T}\sum_{j=1}^{t-1}\alpha_{ji}}$ $(\leq \beta_{t-1,i})$, we have

$$\beta'_{t+1,i} - \beta'_{ti} = \frac{1}{\beta'_{t+1,i} + \beta'_{ti}}\frac{\alpha_{ti}}{\log T} \geq \frac{\alpha_{ti}}{2\beta_{ti}\log T},$$

which implies

$$\sum_{t=1}^{T}\frac{\alpha_{ti}}{\beta_{ti}} \leq 2\log T \cdot \sum_{t=1}^{T}(\beta'_{t+1,i} - \beta'_{ti}) = 2\log T \cdot \beta'_{T+1,i} \leq 2\log T \cdot \beta_{T+1,i}.$$

Combining (20) and (21), and substituting $\gamma = \log T$, we obtain

$$R_T \leq 6\log T \cdot \mathbf{E}\left[\sum_{i=1}^{d}\beta_{T+1,i}\right] + d^2. \tag{22}$$

From this and the definitions of $\alpha_{ti}$ and $\beta_{ti}$ in (4), we have (7) and (8). In fact, as we have

$$\beta_{T+1,i} = \sqrt{2 + \frac{1}{\log T}\sum_{t=1}^{T}\alpha_{ti}} \leq \sqrt{2} + \sqrt{\frac{1}{\log T}\sum_{t=1}^{T}\alpha_{ti}} \tag{23}$$

from (4), by combining this with (22), we obtain

$$R_T \leq 6 \log T \cdot \mathbf{E} \left[ \sum_{i=1}^{d} \left( \sqrt{2} + \sqrt{\frac{1}{\log T} \sum_{t=1}^{T} \alpha_{ti}} \right) \right] + d^2$$

$$\leq 6 \cdot \mathbf{E} \left[ \sum_{i=1}^{d} \left( \sqrt{\log T \sum_{t=1}^{T} \alpha_{ti}} \right) \right] + d^2 + 9d \log T. \tag{24}$$

From this and $\alpha_{ti} \leq a_{ti}(\ell_{ti} - m_{ti})^2$, which immediately follows from the definition of $\alpha_{ti}$ in (4), we have

$$R_T \leq 6 \cdot \mathbf{E} \left[ \sum_{i=1}^{d} \sqrt{\log T \sum_{t=1}^{T} a_{ti}(\ell_{ti} - m_{ti})^2} \right] + d^2 + 9d \log T. \tag{25}$$

Further, as we have $(\ell_{ti} - m_{ti})^2 \leq 1$ and $\mathbf{E}[a_{ti}|x_{ti}] = x_{ti}$, we have

$$\mathbf{E}[\alpha_{ti}] \leq \mathbf{E} \left[ a_{ti} \min \left\{ 1, \frac{1 - x_{ti}}{\gamma x_{ti}^2} \right\} \right] = \mathbf{E} \left[ \min \left\{ x_{ti}, \frac{1 - x_{ti}}{\gamma x_{ti}} \right\} \right] \leq 2\,\mathbf{E} \left[ \min \left\{ x_{ti}, \frac{1 - x_{ti}}{\sqrt{\gamma}} \right\} \right]. \tag{26}$$

In fact, if $\min \left\{ x_{ti}, \frac{1 - x_{ti}}{\sqrt{\gamma}} \right\} = \frac{1 - x_{ti}}{\sqrt{\gamma}}$, we have $x_{ti} \geq \frac{1}{1 + \sqrt{\gamma}} \geq \frac{1}{2\sqrt{\gamma}}$, which implies $\frac{1 - x_{ti}}{\gamma x_{ti}} \leq 2\frac{1 - x_{ti}}{\sqrt{\gamma}}$. Combining (24) and (26), substituting $\gamma = \log T$, and applying Jensen's inequality, we obtain

$$R_T \leq 9 \cdot \sum_{i=1}^{d} \sqrt{\log T\, \mathbf{E} \left[ \sum_{t=1}^{T} \min \left\{ x_{ti}, \frac{1 - x_{ti}}{\sqrt{\log T}} \right\} \right]} + d^2 + 9d \log T. \tag{27}$$

### A.4 Proof of Corollary 1

We start with showing the following lemma:

**Lemma 3.** *If $m_t$ is given by (5), it holds for any $\{u_t\}_{t=1}^{T} \subseteq [0, 1]^d$ that*

$$\sum_{t=1}^{T} a_{ti}(\ell_{ti} - m_{ti})^2 \leq 2 \sum_{t=1}^{T} a_{ti}(\ell_{ti} - u_{ti})^2 + \sum_{t=1}^{T} 16|u_{ti} - u_{t+1,i}| + 1 \tag{28}$$

*for any $i \in [d]$.*

*Proof.* From the definition (5) of $m_t$, we have

$$a_{ti}(\ell_{ti} - m_{ti})^2 - a_{ti}(\ell_{ti} - u_{ti})^2 = a_{ti}(2\ell_{ti} - m_{ti} - u_{ti})(u_{ti} - m_{ti})$$
$$\leq 2a_{ti}(\ell_{ti} - m_{ti})(u_{ti} - m_{ti})$$
$$= 2a_{ti}(\ell_{ti} - m_{ti})(m_{t+1,i} - m_{ti}) + 2a_{ti}(\ell_{ti} - m_{ti})(u_{ti} - m_{t+1,i})$$
$$= \frac{1}{2}a_{ti}(\ell_{ti} - m_{ti})^2 + 8(m_{t+1,i} - m_{ti})(u_{ti} - m_{t+1,i})$$
$$\leq \frac{1}{2}a_{ti}(\ell_{ti} - m_{ti})^2 + 4((u_{ti} - m_{ti})^2 - (u_{ti} - m_{t+1,i})^2),$$

where the third inequality follows from $m_{t+1,i} - m_{ti} = \frac{1}{4}a_{ti}(\ell_{ti} - m_{ti})$. We hence have

$$a_{ti}(\ell_{ti} - m_{ti})^2 \leq 2a_{ti}(\ell_{ti} - u_{ti})^2 + 8((u_{ti} - m_{ti})^2 - (u_{ti} - m_{t+1,i})^2).$$

By taking the summation of this for $t \in [T]$, we obtain

$$\sum_{t=1}^{T} a_{ti}(\ell_{ti} - m_{ti})^2 \leq 2\sum_{t=1}^{T} a_{ti}(\ell_{ti} - u_{ti})^2 + 8\sum_{t=1}^{T}((u_{ti} - m_{ti})^2 - (u_{ti} - m_{t+1,i})^2)$$

$$\leq 2\sum_{t=1}^{T} a_{ti}(\ell_{ti} - u_{ti})^2 + 8\sum_{t=1}^{T-1}((u_{t+1,i} - m_{t+1,i})^2 - (u_{ti} - m_{t+1,i})^2) + (u_{1i} - m_{1i})^2$$

$$\leq 2\sum_{t=1}^{T} a_{ti}(\ell_{ti} - u_{ti})^2 + 8\sum_{t=1}^{T-1}(u_{t+1,i} + u_{ti} - 2m_{t+1,i})(u_{t+1,i} - u_{ti}) + 1$$

$$\leq 2\sum_{t=1}^{T} a_{ti}(\ell_{ti} - u_{ti})^2 + \sum_{t=1}^{T+1} 16|u_{t+1,i} - u_{ti}| + 1.$$

$\square$

Note that $\{u_t\}_{t=1}^{T}$ in this lemma does not appear in the algorithm and is used only in the analysis. Lemma 3 can be seen as a special case of Theorem 11.4 in [16].

*Proof of Corollary 1.* We first show $R_T = O\left(\sqrt{d \log T \, \mathbf{E}\left[\sum_{t=1}^{T} \ell_t^\top a^*\right]} + d^2 + d \log T\right)$. By substituting $u_{ti} = 0$ for all $t \in [T]$ and $i \in [d]$, from (28), we obtain

$$\mathbf{E}\left[\sum_{t=1}^{T}\sum_{i=1}^{d} a_{ti}(\ell_{ti} - m_{ti})^2\right] \leq 2\,\mathbf{E}\left[\sum_{t=1}^{T}\sum_{i=1}^{d} a_{ti}\ell_{ti}^2\right] + 1 \leq 2\,\mathbf{E}\left[\sum_{t=1}^{T} \ell_t^\top a_t\right] + 1$$

$$= 2\left(R_T(a^*) + \mathbf{E}\left[\sum_{t=1}^{T} \ell_t^\top a^*\right]\right) + 1,$$

where the first inequality follows from (28) with $u_{ti} = 0$, the second inequality follows from $\ell_t \in [0,1]^d$ and $a_t \in \{0,1\}^d$, and the last equality follows from the definition of $R_T(a^*)$ in (1). Combining this with (25) and applying Jensen's inequality, we obtain

$$R_T(a^*) \leq 6 \cdot \mathbf{E}\left[\sum_{i=1}^{d}\sqrt{\log T \sum_{t=1}^{T} a_{ti}(\ell_{ti} - m_{ti})^2}\right] + d^2 + 9d\log T$$

$$\leq 6 \cdot \mathbf{E}\left[\sqrt{d\log T \sum_{i=1}^{d}\sum_{t=1}^{T} a_{ti}(\ell_{ti} - m_{ti})^2}\right] + d^2 + 9d\log T$$

$$\leq 6 \cdot \sqrt{d\log T \, \mathbf{E}\left[\sum_{i=1}^{d}\sum_{t=1}^{T} a_{ti}(\ell_{ti} - m_{ti})^2\right]} + d^2 + 9d\log T$$

$$\leq 6 \cdot \sqrt{2d\log T \left(R_T(a^*) + \mathbf{E}\left[\sum_{t=1}^{T} \ell_t^\top a^*\right] + 1\right)} + d^2 + 9d\log T,$$

where the second inequality follows from the Cauchy–Schwarz inequality. By solving the quadratic inequation $(R_T(a^*) - d^2 - 9d\log T)^2 \leq 72d\log T\left(R_T(a^*) + \mathbf{E}\left[\sum_{t=1}^{T} \ell_t^\top a^*\right] + 1\right)$ in $R_T(a^*)$, we obtain $R_T(a^*) = O\left(\sqrt{d\log T \, \mathbf{E}\left[\sum_{t=1}^{T} \ell_t^\top a^*\right]} + d^2 + d\log T\right)$.

Similarly, we can show that $R_T = O\left(\sqrt{d\log T \, \mathbf{E}\left[\sum_{t=1}^{T}\sum_{i=1}^{d} a_{ti}(\ell_{ti} - \bar{\ell}_i)^2\right]} + d^2 + d\log T\right) = O\left(\sqrt{d\log T \, \mathbf{E}\left[\sum_{t=1}^{T} \|\ell_t - \bar{\ell}\|_2^2\right]} + d^2 + d\log T\right)$ by substituting $u_t = \bar{\ell}$ for all $t \in [T]$, to (28).

We can show that $R_T = O\left(\sqrt{d \log T\, \mathbf{E}\left[\sum_{t=1}^{T-1} \|\ell_t - \ell_{t+1}\|_1\right]} + d^2 + d \log T\right)$ as well by substituting $u_t = \ell_t$ for all $t \in [T]$, into (28). $\qquad\square$

## A.5 Proof of Corollary 2

We provide improved regret upper bounds via the following regret *lower* bounds:

**Lemma 4.** *For $a^* \in \mathcal{A}$, denote $I^* = \{i \in [d] \mid a_i^* = 1\}$ and $J^* = [d] \setminus I^*$. In a stochastic regime with adversarial corruptions, for any algorithm, the regret is bounded from below as*

$$R_T(a^*) \geq \frac{\Delta}{B'(\mathcal{A})}\, \mathbf{E}\left[\sum_{t=1}^{T}\left(\sum_{i \in I^*}(1 - a_{ti}) + \sum_{i \in J*} a_{ti}\right)\right] - 2Cm, \tag{29}$$

*where $B'(\mathcal{A}) > 0$ is defined as*

$$B'(\mathcal{A}) = \begin{cases} 2m & (\textit{general cases}) \\ 2\min\{m, d-m\} & (\textit{size-invariant semi-bandits}) \\ 2 & (\textit{matroid semi-bandits}) \end{cases}. \tag{30}$$

*Further, for matroid semi-bandits, we have*

$$R_T(a^*) \geq \frac{1}{2}\, \mathbf{E}\left[\sum_{t=1}^{T}\left(\Delta \sum_{i \in I^*}(1 - a_{ti}) + \sum_{i \in J*} \Delta_i a_{ti}\right)\right] - 2Cm, \tag{31}$$

*where we define $\Delta_i = \min_{a \in \mathcal{A}:a_i=1} \mu^\top a - \mu^\top a^*$.*

*Proof.* Let us recall the conditions in a stochastic regime with adversarial corruptions:

$$\mathbf{E}[\ell_t'] = \mu \quad (t \in [T]), \tag{32}$$

$$\sum_{t=1}^{T} \|\ell_t - \ell_t'\|_\infty \leq C, \tag{33}$$

$$a^* \in \arg\min_{a \in \mathcal{A}} \mu^\top a, \tag{34}$$

$$\Delta = \min_{a \in \mathcal{A}\setminus\{a^*\}} \mu^\top a - \mu^\top a^* > 0. \tag{35}$$

From these conditions, we have

$$\begin{aligned}
R_T(a^*) = \mathbf{E}\left[\sum_{t=1}^{T} \ell_t^\top(a_t - a^*)\right] &= \mathbf{E}\left[\sum_{t=1}^{T} \ell_t'^\top(a_t - a^*) + \sum_{t=1}^{T}(\ell_t - \ell_t')^\top(a_t - a^*)\right] \\
&\geq \mathbf{E}\left[\sum_{t=1}^{T} \mu^\top(a_t - a^*) - \sum_{t=1}^{T} \|\ell_t - \ell_t'\|_\infty \|a_t - a^*\|_1\right] \\
&\geq \mathbf{E}\left[\sum_{t=1}^{T} \mu^\top(a_t - a^*) - 2m \sum_{t=1}^{T} \|\ell_t - \ell_t'\|_\infty\right] \\
&\geq \mathbf{E}\left[\sum_{t=1}^{T} \mu^\top(a_t - a^*)\right] - 2Cm \geq \mathbf{E}\left[\sum_{t=1}^{T} \Delta \cdot \mathbf{1}[a_t \neq a^*]\right] - 2Cm, \tag{36}
\end{aligned}$$

where the first, the third and the last inequality follows from (32), (33) and (35), respectively, and the second inequality follows from $\|a\|_1 \leq m$ for all $a \in \mathcal{A}$. As we have $\|a_t - a^*\|_1 \leq 2m$ for any $a_t \in \mathcal{A}$, we have

$$\mathbf{1}[a_t \neq a^*] \geq \frac{1}{2m}\|a_t - a^*\|_1 = \frac{1}{2m}\left(\sum_{i \in I^*}(1 - a_{ti}) + \sum_{i \in J*} a_{ti}\right). \tag{37}$$

Combining this with (36), we obtain (29) in which $\mathcal{B}'(\mathcal{A}) = 2m$. Similarly for the case of size-invariant semi-bandits, as we have $\|a_t - a^*\|_1 \leq 2\min\{m, d - m\}$ for any $a_t$, we obtain (29) in which $\mathcal{B}'(\mathcal{A}) = 2\min\{m, d - m\}$.

For the matroid case, we have

$$\mu^\top(a - a^*) \geq \sum_{i \in J^*} \Delta_i a_i \tag{38}$$

for any $a \in \mathcal{A}$. This can be shown via the symmetric basis-exchange property of matroid bases [10]. Denote $I = \{i \in [d] \mid a_i = 1\}$ and $k = |I \setminus I^*|$. We consider the following sequence of bases $\{I_j\}_{j=0}^k$:

- Set $I_0 = I^*$.

- For $j = 0, 1, \ldots, k - 1$: choose $i_j \in I_j \setminus I^*$ arbitrarily. From the symmetric basis-exchange property, there exists $i'_j \in I^* \setminus I_j$ such that both $(I_j \cup i'_j) \setminus \{i_j\}$ and $(I^* \cup i_j) \setminus \{i'_j\}$ are bases. Let $I_{j+1} = (I_j \cup i'_j) \setminus \{i_j\}$.

As we have $|I_{j+1} \setminus I^*| = |I_j \setminus I^*| - 1$ for $j \in [k - 1]$, we have $|I_k \setminus I^*| = 0$, and consequently, $I_k = I^*$ holds. Similarly, we can see that $\{i_j\}_{j=1}^k = I \setminus I^*$. The value of $\mu^\top(a - a^*)$ can be expressed as

$$\mu^\top(a - a^*) = \sum_{j=1}^k (\mu_{i_j} - \mu_{i'_j}) \geq \sum_{j=1}^k \Delta_{i_j} = \sum_{i \in I \setminus I^*} \Delta_i = \sum_{i \in J^*} \Delta_i a_i, \tag{39}$$

where the inequality follows from the definition of $\Delta_i$ and the fact that $(I^* \cup i_j) \setminus \{i'_j\}$ is a base. From this, $\Delta = \min_{i \in J^*} \Delta_i$, and the fact that $\sum_{i \in J^*} a_i = \sum_{i \in I^*}(1 - a_i)$, we have

$$\mu^\top(a - a^*) \geq \frac{\Delta}{2} \sum_{i \in J^*} a_i + \frac{1}{2} \sum_{i \in J^*} \Delta_i a_i = \frac{\Delta}{2} \sum_{i \in I^*}(1 - a_i) + \frac{1}{2} \sum_{i \in J^*} \Delta_i a_i. \tag{40}$$

Combining this with (36), we obtain (31). Further, it follows from $\Delta \leq \Delta_i$ for any $i \in J^*$ that (29) with $\mathcal{B}'(\mathcal{A}) = 2$ holds. $\qquad\square$

*Proof of Corollary 2.* From (27) and (29), for any $\lambda > 0$, we have

$$R_T(a^*) = (1 + \lambda)R_T(a^*) - \lambda R_T(a^*)$$

$$\leq (1 + \lambda)\left(9 \cdot \sum_{i=1}^d \sqrt{\log T \, \mathbf{E}\left[\sum_{t=1}^T \min\left\{x_{ti}, \frac{1 - x_{ti}}{\sqrt{\log T}}\right\}\right]} + d^2 + 9d\log T\right)$$

$$- \lambda\left(\frac{\Delta}{\mathcal{B}'(\mathcal{A})} \mathbf{E}\left[\sum_{t=1}^T \left(\sum_{i \in I^*}(1 - a_{ti}) + \sum_{i \in J^*} a_{ti}\right)\right] - 2Cm\right)$$

$$\leq \sum_{i \in J^*}\left(9(1 + \lambda)\sqrt{\log T}\sqrt{\mathbf{E}\left[\sum_{t=1}^T x_{ti}\right]} - \frac{\lambda\Delta}{\mathcal{B}'(\mathcal{A})}\mathbf{E}\left[\sum_{t=1}^T x_{ti}\right]\right)$$

$$+ \sum_{i \in I^*}\left(9(1 + \lambda)(\log T)^{1/4}\sqrt{\mathbf{E}\left[\sum_{t=1}^T(1 - x_{ti})\right]} - \frac{\lambda\Delta}{\mathcal{B}'(\mathcal{A})}\mathbf{E}\left[\sum_{t=1}^T(1 - x_{ti})\right]\right)$$

$$+ (1 + \lambda)(d^2 + 9d\log T) + 2\lambda Cm$$

$$\leq \sum_{i \in J^*} \frac{81(1 + \lambda)^2 \mathcal{B}'(\mathcal{A})\log T}{4\lambda\Delta} + \sum_{i \in I^*} \frac{81(1 + \lambda)^2 \mathcal{B}'(\mathcal{A})\sqrt{\log T}}{4\lambda\Delta}$$

$$+ (1 + \lambda)(d^2 + 9d\log T) + 2\lambda Cm,$$

$$= \left(|J^*| + |I^*|\frac{1}{\sqrt{\log T}}\right)\frac{81(1 + \lambda)^2 \mathcal{B}'(\mathcal{A})\log T}{4\lambda\Delta} + (1 + \lambda)(d^2 + 9d\log T) + 2\lambda Cm,$$

where the first inequality follows from (27) and (29), the second inequality follows from $\mathbf{E}[a_t|x_t] = x_t$, and the third inequality follows from the inequality $a\sqrt{x} - bx = -b(\sqrt{x} - \frac{a}{2b})^2 + \frac{a^2}{4b} \le \frac{a^2}{4b}$ that holds for any $x \ge 0$, $a \ge 0$, and $b > 0$. By setting $B(\mathcal{A}) = \frac{1}{2}\left(|J^*| + |I^*|\frac{1}{\sqrt{\log T}}\right)B'(\mathcal{A})$, we have

$$R_T(a^*) \le \frac{81(1+\lambda)^2 B(\mathcal{A})\log T}{2\lambda\Delta} + (1+\lambda)(d^2 + 9d\log T) + 2\lambda Cm$$

$$\le \frac{81B(\mathcal{A})\log T}{\Delta} + d^2 + 9d\log T + \lambda\left(\frac{81B(\mathcal{A})\log T}{2\Delta} + d^2 + 9d\log T + Cm\right) + \frac{1}{\lambda}\frac{81B(\mathcal{A})\log T}{2\Delta}.$$

By choosing $\lambda = \Theta\left(\sqrt{\left(\frac{B(\mathcal{A})\log T}{\Delta}\right) / \left(\frac{B(\mathcal{A})\log T}{\Delta} + d^2 + d\log T + Cm\right)}\right)$, we obtain $R_T(a^*) = O\left(\frac{B(\mathcal{A})\log T}{\Delta} + \sqrt{B(\mathcal{A})\frac{Cm\log T}{\Delta}} + d^2\right)$. From conditions in (30) and the definition of $B(\mathcal{A})$, we can confirm that (9) holds. $\square$

The regret bound in Remark 4 can be similarly shown, by combining (27) and (31).

## A.6 Proof of Theorem 3

Combining (10) and Theorem 2, we obtain

$$R_T \le D \cdot \mathbf{E}\left[\sqrt{2\sum_{t=1}^{T}((\ell_t - u_t)^\top a_t)^2 + 8p'\sum_{t=1}^{T-1}\|u_t - u_{t+1}\|_q + 8p'}\right] \tag{41}$$

for any $\{u_t\}_{t=1}^{T} \subseteq \mathcal{L}$. Considering a special case of $u_t = \bar{\ell}$ for all $t \in [T]$, we have

$$R_T \le D \cdot \mathbf{E}\left[\sqrt{2\sum_{t=1}^{T}((\ell_t - \bar{\ell})^\top a_t)^2 + 8p'}\right] \le D \cdot \mathbf{E}\left[\sqrt{2\sum_{t=1}^{T}\|\ell_t - \bar{\ell}\|_q^2\|a_t\|_p^2 + 8p'}\right]$$

$$\le D \cdot \mathbf{E}\left[\sqrt{2\sum_{t=1}^{T}\|\ell_t - \bar{\ell}\|_q^2 + 8p'}\right] = D \cdot \mathbf{E}\left[\sqrt{2Q_q + 8p'}\right].$$

Similarly, by considering a special case of $u_t = \ell_t$ for all $t \in [T]$, we obtain

$$R_T \le D \cdot \mathbf{E}\left[\sqrt{8p'\sum_{t=1}^{T-1}\|\ell_t - \ell_{t+1}\|_q + 8p'}\right] = D \cdot \mathbf{E}\left[\sqrt{8p'V_q + 8p'}\right].$$

Further, if $\ell_t^\top a_t \in [0,1]$ for all $t \in [T]$, by substituting $u_t = 0$ for all $t \in [T]$, we obtain

$$R_T(a^*) \le D \cdot \mathbf{E}\left[\sqrt{2\sum_{t=1}^{T}(\ell_t^\top a_t)^2 + 8p'}\right] \le D \cdot \mathbf{E}\left[\sqrt{2\sum_{t=1}^{T}\ell_t^\top a_t + 8p'}\right]$$

$$\le D \cdot \sqrt{2\left(R_T(a^*) + \mathbf{E}\left[\sum_{t=1}^{T}\ell_t^\top a*\right]\right) + 8p'} = D \cdot \sqrt{2\left(R_T(a^*) + \mathbf{E}[L^*]\right) + 8p'},$$

where the second inequality follows from $\ell_t^\top a_t \in [0,1]$ and the second inequality follows the definition of $R_T(a^*)$ and Jensen's inequality. This implies $(R_T(a^*))^2 \le D^2(2(R_T(a^*) + \mathbf{E}[L^*]) + 8p')$. By solving this quadratic inequation in $R_T(a^*)$, we obtain $R_T(a^*) = O(D \cdot (\sqrt{\mathbf{E}[L^*]} + p'))$. $\square$

## B Regret Bounds for an Existing Method in Corrupted Stochastic Settings

In this section, we see that the algorithm by Zimmert et al. [71] achieves a regret bound of

$$R_T(a^*) = O\left(\frac{m}{\Delta}(d\log T + m) + \sqrt{\frac{Cm^2}{\Delta}(d\log T + m)}\right) \tag{42}$$

in stochastic regimes with adversarial corruptions.

## B.1 Natation and known results

From (4) in the paper [71], the regret for their algorithm is bounded as

$$R_T(a^*) \leq \sum_{t=1}^{T} \frac{25}{\sqrt{t}} \left( f(\mathbf{E}[a_t]) + g(\mathbf{E}[a_t]) \right) + c, \tag{43}$$

where $f$, $g$ and $c$ are defined by

$$f(x) = \sum_{i \in J^*} \sqrt{x_i}, \quad g(x) = \sum_{i \in I^*} (\gamma^{-1} - \gamma \log(1 - x_i))(1 - x_i), \quad c = \frac{58m}{\gamma^2} \tag{44}$$

and $\gamma \in (0, 1]$ is an input parameter. Note that $I^*$ and $J^*$ are defined in the same way as in Lemma 4. Define $\Delta_x$ for $x \in \mathcal{A}$, $r(\cdot)$, and $P(\cdot)$ in the same way as in the paper by Zimmert et al. [71]. Similarly, we further define $C_{sto}$ and $C_{add}(u)$ by

$$C_{sto} := \max_{\alpha \in [0,\infty]^{\mathcal{A}}} (f(\bar{\alpha}) - r(\alpha)), \quad C_{add}(u) := \sum_{t=1}^{\infty} \max_{\alpha \in \Delta(\mathcal{A})} \left( \frac{u}{\sqrt{t}} f(\bar{\alpha}) - r(\alpha) \right) \tag{45}$$

for any $u > 0$, where $\bar{\alpha} \in \mathbb{R}^d$ is defined by $\bar{\alpha} = \sum_{x \in \mathcal{A}} \alpha_x x$. As can be seen from Section A.3 of [71], $C_{sto}$ and $C_{add}$ are bounded as

$$C_{sto} = O\left(\frac{md}{\Delta}\right), \quad C_{add}(u) = O\left(\frac{m^2 u^2}{\gamma^2 \Delta}\right) \tag{46}$$

in general. We note that $\Delta$ in this paper corresponds to $\Delta_{\min}$ in [71] and that Zimmert et al. [71] have provided even better bounds for $C_{sto}$ and $C_{add}(u)$ in some special cases including problems with full combinatorial set or $m$-set. We only consider the case in which $C_{sto}$ and $C_{add}(u)$ are bounded as in (46) for the sake of simplicity.

## B.2 Regret analysis in stochastic regime with adversarial corruptions

In stochastic regimes with adversarial corruptions, the regret is bounded from below as follows:

$$R_T(a^*) \geq \sum_{t=1}^{T} r(P(\mathbf{E}[a_t])) - 2Cm, \tag{47}$$

which can be shown in a similar way to Lemma 4. Combining (43) with (47), for any $\lambda > 0$, we obtain

$$R_T(a^*) = (1 + \lambda)R_T(a^*) - \lambda R_T(a^*)$$

$$\leq (1 + \lambda) \left( \sum_{t=1}^{T} \frac{25}{\sqrt{t}} \left( f(\mathbf{E}[a_t]) + g(\mathbf{E}[a_t]) \right) + c \right) - \lambda \left( \sum_{t=1}^{T} r(P(\mathbf{E}[a_t])) - 2Cm \right)$$

$$= \sum_{t=1}^{T} \left( \frac{25(1+\lambda)}{\sqrt{t}} f(\mathbf{E}[a_t]) - \frac{\lambda}{2} r(P(\mathbf{E}[a_t])) \right) + \sum_{t=1}^{T} \left( \frac{50(1+\lambda)}{\sqrt{t}} g(\mathbf{E}[a_t]) - \frac{\lambda}{2} r(P(\mathbf{E}[a_t])) \right)$$

$$+ (1 + \lambda)c + 2\lambda Cm. \tag{48}$$

Each term in the right-hand side can be bounded via similar arguments to in the proof of Theorem 1 by Zimmert et al. [71]. In fact, we have

$$\sum_{t=1}^{T} \left( \frac{25(1+\lambda)}{\sqrt{t}} f(\mathbf{E}[a_t]) - \frac{\lambda}{2} r(P(\mathbf{E}[a_t])) \right) \leq \sum_{t=1}^{T} \max_{\alpha \in \Delta(\mathcal{A})} \left( \frac{25(1+\lambda)}{\sqrt{t}} f(\bar{\alpha}) - \frac{\lambda}{2} r(\alpha) \right)$$

$$\leq \sum_{t=1}^{T} \max_{\alpha \in [0,\infty]^{\mathcal{A}}} \left( \frac{25(1+\lambda)}{\sqrt{t}} f\left( \frac{50^2(1+\lambda)^2}{\lambda^2 t} \bar{\alpha} \right) - \frac{\lambda}{2} r\left( \frac{50^2(1+\lambda)^2}{\lambda^2 t} \alpha \right) \right)$$

$$= \sum_{t=1}^{T} \frac{50^2(1+\lambda^2)}{2\lambda t} \max_{\alpha \in [0,\infty]^{\mathcal{A}}} (f(\bar{\alpha}) - r(\alpha)) = O\left( \frac{(1+\lambda)^2}{\lambda} C_{sto} \log T \right)$$

$$= O\left( \left( \lambda + \frac{1}{\lambda} \right) C_{sto} \log T \right), \tag{49}$$

where the first equality follows from the fact that $r$ is linear and that $f(ux) = \sqrt{u}f(x)$ holds for any scalar $u \geq 0$, and the second equality follows from (46). Similarly, we can see that

$$\sum_{t=1}^{T} \left( \frac{50(1+\lambda)}{\sqrt{t}} g(\mathbf{E}[a_t]) - \frac{\lambda}{2} r(P(\mathbf{E}[a_t])) \right) = \frac{\lambda}{2} \sum_{t=1}^{T} \left( \frac{100(1+\lambda)}{\lambda\sqrt{t}} g(\mathbf{E}[a_t]) - r(P(\mathbf{E}[a_t])) \right)$$

$$\leq \frac{\lambda}{2} C_{add} \left( \frac{100(1+\lambda)}{\lambda\sqrt{t}} \right) = O\left( \frac{\lambda}{2} \left( \frac{1+\lambda}{\lambda^2} \right)^2 \frac{m^2}{\gamma^2\Delta} \right) = O\left( \left( \lambda + \frac{1}{\lambda} \right) \frac{m^2}{\gamma^2\Delta} \right), \tag{50}$$

where the inequality follows for the definition of $C_{add}(u)$ in (45) and the second inequality follows from (46). Combining (48), (49) and (50), we obtain

$$R_T(a^*) = O\left( \lambda \left( C_{sto} \log T + \frac{m^2}{\gamma^2\Delta} + Cm + c \right) + \frac{1}{\lambda} \left( C_{sto} \log T + \frac{m^2}{\gamma^2\Delta} \right) + c \right).$$

By choosing $\lambda = \sqrt{\frac{C_{sto} \log T + m^2/\gamma^2\Delta}{C_{sto} \log T + m^2/\gamma^2\Delta + Cm + c}}$, we obtain

$$R_T(a^*) = O\left( \sqrt{\left( C_{sto} \log T + \frac{m^2}{\gamma^2\Delta} + Cm + c \right) \left( C_{sto} \log T + \frac{m^2}{\gamma^2\Delta} \right)} + c \right)$$

$$= O\left( C_{sto} \log T + \frac{m^2}{\gamma^2\Delta} + c + \sqrt{(Cm + c)\left( C_{sto} \log T + \frac{m^2}{\gamma^2\Delta} \right)} \right)$$

$$= O\left( \frac{dm}{\Delta} \log T + \frac{m^2}{\gamma^2\Delta} + c + \sqrt{(Cm + c)\left( \frac{dm}{\Delta} \log T + \frac{m^2}{\gamma^2\Delta} \right)} \right),$$

where the last equality follows from (46). If we set $\gamma = 1$, we have $c = O(m)$ and hence the regret is bounded as

$$R_T = O\left( \frac{m}{\Delta}(d \log T + m) + \sqrt{\frac{Cm^2}{\Delta}(d \log T + m)} \right),$$

which means that (42) holds.