# OpenReview forum: "Hybrid Regret Bounds for Combinatorial Semi-Bandits and Adversarial Linear Bandits"
_NeurIPS.cc/2021/Conference — NeurIPS 2021 Poster_

### Official Review · Reviewer_Q7uf · 2021-07-09

**Rating:** 6
**Confidence:** 3

**Summary:**

This paper studies the hybrid regret bounds for combinatorial semi-bandits (in stochastic, stochastic with corruption and adversarial settings simultaneously) and linear bandits (w.r.t. three different parameters). The paper improves the previous results by combining optimistic predictors with FTRL and the alg. of tracking the best linear predictor.

**Limitations And Societal Impact:**

Yes.

**Main Review:**

Originality: The task has been studied before and this work proposes a new method to improve the previous results. It might seem that the proposed method is a direct combination of known results. The contribution of this work highly depends on how difficult it is to combine the known methods. It would be important to address the main challenges of combining these methods and characterize the novelty.

Quality: All claims are well supported.

Clarity: The submission lacks discussion and explanation.  In section 4 and 5, it seems that the paper just lists the algorithms and the quantities without much explanation. For example, when comparing the proposed algorithm with the existing one, the paper does not explain why these changes improve the results. When discussing the reason for using a log-barrier regularizer instead of Tsallis entropy (Line 273-278), it is hard to understand it without checking other works.

Significance: The significance of this work is not obvious to me.

-------------------------------------------------------------------------------------
The authors addressed my concerns about the technical novelty of this submission. I am happy to raise my score. I hope the authors could add more discussion in Sec 4 and 5 and address the difference between the algorithms in this work and those in previous work.

**Time Spent Reviewing:**

4.5

---

> ### Author Response · Authors · 2021-08-09
> **Response to Reviewer Q7uf**
>
> Thank you for all the useful comments.
> We hope the following response addresses your concerns.
>
> > It might seem that the proposed method is a direct combination of known results. The contribution of this work highly depends on how difficult it is to combine the known methods. It would be important to address the main challenges of combining these methods and characterize the novelty.
>
> We believe that the proposed algorithm is not an immediate result to follow from existing works,
> although it is based on a combination of existing techniques.
> In fact,
> the combination of log-barrier regularization,
> hybrid regularization and entry-wise adaptive learning rates requires non-trivial considerations.
> For example,
> the update rule for the learning rate in this work is novel,
> as we claims in the response to Reviewer Ny32.
>
>
> > The submission lacks discussion and explanation. In section 4 and 5, it seems that the paper just lists the algorithms and the quantities without much explanation. For example, when comparing the proposed algorithm with the existing one, the paper does not explain why these changes improve the results.
>
> Lines 92-107 in the introduction explain how each change and technique improves the results.
> As mentioned in line 92-107,
> we use an optimistic-prediction framework to obtain data-dependent regret bounds.
> To employ the optimistic-prediction framework,
> the proposed algorithm uses log-barrier regularization instead of Tsallis entropy (
> So far, it is not known if the combination of Tsallis entropy and optimistic prediction works well
> ).
> The entry-wise adaptive learning rates are required to obtain the best-of-both-worlds property.
> In the revised version, we will add a similar explanation and description of the parameter interpretation in Sections 4 and 5.
>
>
> > When discussing the reason for using a log-barrier regularizer instead of Tsallis entropy (Line 273-278), it is hard to understand it without checking other works.
>
> The reason for using a log-barrier regularizer are discussed in Lines 102-104.
> Briefly speaking,
> the main reason is that we could not show that algorithms using Tsallis entropy regularization combined with optimistic prediction works well.
> In fact,
> if we use optimistic prediction,
> we cannot use Lemma 5 in [Zimmert et al., 2019] as the assumption of $-1 \leq \hat{\ell}_i$ in the lemma does not hold,
> and consequently,
> existing analysis methods for Tsallis entropy regularization do not directly apply.
>
> [Zimmert et al., 2019] J. Zimmert, H. Luo, and C.-Y. Wei. Beating stochastic and adversarial semi-bandits optimally and simultaneously. In International Conference on Machine Learning, pages 7683–7692. PMLR, 2019.

---

### Official Review · Reviewer_Ny32 · 2021-07-12

**Rating:** 7
**Confidence:** 4

**Summary:**

This paper proposes an algorithm that obtains multiple data-dependent regret bounds for semi-bandits simultaneously with optimal regret bounds in the stochastic and intermediate regimes.

Additionally they propose an adversarial linear bandit algorithm that obtains several problem dependent regret bounds simultaneously such as optimal path length bounds.

**Limitations And Societal Impact:**

Yes.

**Main Review:**

This paper extends recent results for the K-armed bandit setting derived in the COLT2021 submission:
"Parameter-Free Multi-Armed Bandit Algorithms with Hybrid Data-Dependent Regret Bounds", to combinatorial semi-bandits.
It combines the same base regularizer (log-barrier) with optimistic FTRL as in Ito 2021, together with the hybrid regularization as in Zimmert et al for semi-bandits.
The paper is well written and the results seem fine.
It recovers the optimal stochastic and adversarial guarantees, as well as a multitude of improved problem dependent regret bounds.

While I think the results are useful and especially the linear bandit results are novel, I am missing a discussion about the relationship to the paper mentioned above. It is unclear to me how much genuine novelty the adaptation to semi-bandit brings compared to previous work.
The paper also lacks a justification for the hybrid regularization. In Zimmert et al., this was introduced mainly to deal with the $m\gg d/2$ case, which the authors do not explicitly care about here. Since the hybrid part adds some extra computational complexity, especially in more general action sets, there should be a strong technical necessity to keeping it.

Minor:
The authors claim that it is unknown if Zimmert et al. also obtains $\sqrt{C}$ bounds. Since they use the same self-bounding regret technique as this paper here and previous papers, it is quite obvious that they do. (The paper is simply older than when the $\sqrt{C}$ bounds first appeared.)

**Time Spent Reviewing:**

3

---

> ### Author Response · Authors · 2021-08-09
> **Response to Reviewer Ny32**
>
> Thank you for all the helpful comments.
> We hope the following response addresses your concerns.
>
> > While I think the results are useful and especially the linear bandit results are novel, I am missing a discussion about the relationship to the paper mentioned above. It is unclear to me how much genuine novelty the adaptation to semi-bandit brings compared to previous work.
>
> Given the results in the COLT2021 submission for the K-armed bandit,
> one novelty is that our proposed algorithm combines hybrid regularization and entry-wise adaptive learning rates.
> The combination of these two techniques requires non-trivial considerations.
> For example,
> the update rule for the learning rate is not just a straightforward extension of existing methods.
> Indeed,
> the definition of $\alpha_{ti}$ in our paper does not coincide with $\nu_{i}^t$ in the COLT2021 paper,
> even in the limit when $\gamma$ approaches $0$ (i.e., when the regularizers are close to standard log-barriers).
> This change is essential in order to obtain the best-of-both-worlds property using the hybrid regularizers,
> the necessity of which is mentioned in the following reply.
> The revised manuscript will cite the COLT 2021 submission and include a comparison with it.
>
> > The paper also lacks a justification for the hybrid regularization.
>
> In the revised manuscript,
> we will add more explanation of the hybrid regularization.
> In order to show best-of-both-worlds regret bounds,
> it is required that
> round-wise regret bounds (e.g., the stability term in [Zimmert et al. 2019])
> converge to $0$ when $x_t$ approaches extreme points in $\\{ 0,  1 \\}^d$.
> As shown in Lemma 2,
> the round-wise regret bounds of our algorithm can be expressed as $O(\sum_{i=1}^d \frac{\alpha_{ti}}{\beta_{ti}} ) =
> O( \sum_{i=1}^d \frac{x_{ti}}{\beta_{ti}} \min \\{ 1, \frac{1 - x_{ti}}{\gamma {x_{ti}^2} }  \\})$ in expectation.
> Without hybrid regularization, i.e., if $\gamma = 0$,
> we cannot obtain the above-mentioned convergence property,
> especially when $x_{ti}$ approaches $1$ for multiple $i$'s.
> Thanks to hybrid regularization (with $\gamma > 0$),
> we can show that the round-wise regret converge to $0$ when $x_t$ approaches *any* points in $\\{ 0, 1 \\}^d$.
>
>
> > Minor: The authors claim that it is unknown if Zimmert et al. also obtains $\sqrt{C}$ bounds. Since they use the same self-bounding regret technique as this paper here and previous papers, it is quite obvious that they do. (The paper is simply older than when the $\sqrt{C}$ bounds first appeared.)
>
> Now that we have been able to show that the algorithm by Zimmert et al. (2019) achieves $O (\log T + \sqrt{C \log T})$-regret,
> we will add this fact and its proof in the revised version.
> We believe that this fact is not entirely trivial as the regret bound by Zimmert et al. (2019) includes a term of $O(\sum_{t=1}^T \frac{1}{\sqrt{t}} \sum_{i:a_i^* = 1} (\gamma^{-1} - \gamma \log (1- x_{ti}))(1-x_{ti}))$ in contrast to Tsallis-INF by Zimmert and Seldin (2021).
> Due to this additional term, the self-bounding analysis in Zimmert and Seldin (2021) does not seem to be directly applicable, and at the time of submission we could not prove an $O(\log T + \sqrt{C \log T})$-bound.
>
>
> [Zimmert and Seldin, 2021] Tsallis-inf: An optimal algorithm for stochastic and adversarial bandits. Journal of Machine Learning Research, 22(28):1–49, 2021.
>
> [Zimmert et al., 2019] J. Zimmert, H. Luo, and C.-Y. Wei. Beating stochastic and adversarial semi-bandits optimally and simultaneously. In International Conference on Machine Learning, pages 7683–7692. PMLR, 2019.

---

### Official Review · Reviewer_JE8q · 2021-07-15

**Rating:** 7
**Confidence:** 3

**Summary:**

This paper proposes an optimistic FTRL-based algorithm that achieves the near-optimal regret for semi-bandits simultaneously in terms of a variety of path-length(variation) instance in adversarial setting, gap in stochastic setting and corruption in the intermediate setting. It also achieves the near-optimal regret for linear-bandits simultaneously in terms of path-length(variation) in adversarial setting,

**Limitations And Societal Impact:**

As I wrote above, I would like authors to discuss more on the influence if not choosing Tsallis entropy with power 1/2.

**Main Review:**

--Originality:
This work is based on the classical optimistic FTRL framework with a hybrid self-bounding regularizer. Unlike the previous works which mostly use doubling tricks or simply time-varying learning rate, It proposes more adaptive, entry-wise, learning-rate type parameters with rigorous analysis to achieve the best of many worlds. I think such modifications are novel.

But the main limitation of novelty is that most initial-stage proof steps (Lemma 1, Lemma 2) are using the standard optimistic FTRL techniques.  And given [Zimmert et al. 2019], [Wei and Luo, 2018], this hybrid log-barrier idea is not very surprising to me.

--Quality:
This submission is technically sound, all the claims are supported.

But one concern to me is that it seems by using a log-barrier type regularizer instead of Tsallis entropy with power 1/2 regularizer, the authors are unable to get a finer regret. (In [Zimmert et al. 2019], they have $\sum_i 1/\Delta_i$ for the full-combinatorial set and fixed-size set, where here it seems to be $m/\Delta_{min}$. If I am correct, I think the author should mention this difference.

--Clarity:
The paper is overall well-written. There are several suggestions:1\ In line 271, it says $\gamma_{ti}$ which I believe should be $\beta_{ti}$ 2\It would be better if you could give some intuitive explanations on $\beta_{ti}$, something like how the two terms in $\alpha_{ti}$ directly connect to equation (7) (8).

--Significance:
I think the best-of-many worlds results are important and the entry-wise adaptive learning rate idea is inspiring.

**Time Spent Reviewing:**

6

---

> ### Author Response · Authors · 2021-08-09
> **Response to Reviewer JE8q**
>
> Thank you for all the useful comments.
> We hope the following response addresses your concerns.
>
>
> > But one concern to me is that it seems by using a log-barrier type regularizer instead of Tsallis entropy with power 1/2 regularizer, the authors are unable to get a finer regret. (In [Zimmert et al. 2019], they have $\sum_{i} 1/ \Delta_i$ for the full-combinatorial set and fixed-size set, where here it seems to be $m / \Delta_{\min}$. If I am correct, I think the author should mention this difference.
>
> Our algorithm achieves $\sum_{i} 1 / \Delta_i$ for the fixed-size set as well as the algorithm by [Zimmert et al. 2019].
> The fixed-size constraint addressed in [Zimmert et al. 2019] is a special case of matroid constraints (as discussed in [Kveton et al. 2014] as well).
> In fact,
> fixed-size constraints (m-set in [Zimmert et al. 2019]) can be represented by uniform matroids as mentioned in Remark 3.
> For matroid semi-bandits,
> Corollary 2 and Remark 4 in our paper imply a finer regret as well as [Zimmert et al. 2019].
>
> The problems over full-combinatorial sets can be reduced to matroid semi-bandits as well.
> For a given full-combinatorial action set $\mathcal{A} = \\{ 0, 1 \\}^d$,
> we define an alternative action set by $\\mathcal{A}' = \\{ a'  \in  \\{ 0, 1 \\}^{2d} \mid \\| a' \\|_1 = d \\}$,
> which forms bases of a uniform matroid over $[2d]$.
>
> Further,
> for the original loss vector $\ell_t \in [0, 1]^d$,
> we define $\ell_t' \in [0, 1]^{2d}$ by $\ell_t' = [ \ell_t^\top, \mathbf{0}^\top ]^\top$.
> Then the problem for $\mathcal{A}$ and $\\{ \ell_t \\}^T_{t=1}$ is equivalent to the problem for $\mathcal{A}'$ and $\\{ \ell_t' \\}^T_{t=1}$.
> Hence,
> we can apply the finer regret bounds for matroid semi-bandits in Corollary 2 and Remark 4.
> In the revised version,
> we will mention these facts more explicitly.
>
>
> > There are several suggestions:1\ In line 271, it says $\gamma_{ti}$ which I believe should be $\beta_{ti}$
>
> Thanks for this comment.
> As the reviewer suggests, $\beta_{ti}$ is correct.
>
> > It would be better if you could give some intuitive explanations on $\beta_{ti}$, something like how the two terms in $\alpha_{ti}$ directly connect to equation (7) (8).
>
> We will add an explanation of the interpretation of the definition on these parameters and how it relates to the regret bound in the revised version.
>
> > As I wrote above, I would like authors to discuss more on the influence if not choosing Tsallis entropy with power 1/2.
>
> Even in the cases of the full-combinatorial set and of the fixed-size set, the proposed algorithm is no worse than the algorithm with Tsallis entropy,
> as we claimed in our response above.
>
> [Kveton et al. 2014] B. Kveton, Z. Wen, A. Ashkan, H. Eydgahi, and B. Eriksson. Matroid bandits: Fast combinatorial optimization with learning. In Conference on Uncertainty in Artificial Intelligence, pages 420–429, 2014.

---

### Official Review · Reviewer_CeRv · 2021-07-16

**Rating:** 7
**Confidence:** 4

**Summary:**

This paper provides an algorithm for the combinatorial semi-bandits that achieves regret bound with three well known notions of data dependence regret in adversary regime. Besides that the provided algorithms attains best of both worlds result (Zimmert et al., 2019) up to a logarithmic factor in adversary setting. Their algorithm in stochastic regime with adversarial corruption achieves $O(\log T + \sqrt{C\log T})$ regret with respect to $T$ as the time horizon and $C$ as the corruption budget. Furthermore, they improved existing regret bounds for adversarial linear bandits to achieve path-length regret bound which in worst case is tight up to a logarithmic factor.

**Limitations And Societal Impact:**

__Limitations:__


- The paper in stochastic and stochastic with adversarial corruption regimes implicitly uses the uniqueness of the best action. This is needed for the self-bounding analysis and must be mentioned as a clear assumption.

- There is a small issue about using “best of three worlds” term. The best of one world result implies there is no better result than that and in other words there must be a matching lower bound for the result. However, obviously the stated result for stochastic with adversarial corruption is not minimax optimal (consider $C \geq T/ \log T$) so in this world the achieved bound seems a rather  good bound but is not the best.


**Main Review:**

## Semi-bandit Part of the Paper
 As it is mentioned in the paper, the key idea of provided algorithm for combinatorial semi-bandits is to combine the zero powered of Tsallis algorithm provided by Zimmert et al. (2019, 2021)  to _optimistic prediction for the losses_ in order to get the best of both world result from the former and data dependent from the later approach.

__Significance and Originality:__

This idea seems interesting and elegant to me. Also circumventing the challenges for the analysis is also a good contribution. However, I have one important issue with the mentioned claims about improvements over Zimmert et al. (2019).
 - Result of the paper for stochastic with adversarial corruption is $O(\log T + \sqrt{C\log T})$  whereas the authors claim Zimmert et al., (2019) , with some straightforward modification of their proof, achieve $O(\log T + C)$. By contrast with the same self-bounding analysis provided by Tsallis-INF paper (Zimmert and Seldin, 2021) , where this paper uses the same analysis approach, it is quite easy to show that  Zimmert et al., (2019) algorithm in this intermediate regime obtains $O(\log T + \sqrt{C\log T})$. The authors need to be more transparent about this point since they are also using the same self-bounding technique for obtaining $O(\log T + \sqrt{C\log T})$ bound. In general there is a work by Masoudian and Seldin (2021)  titled: "Improved Analysis of Robustness of the Tsallis-INF Algorithm to Adversarial Corruptions in Stochastic Multiarmed Bandits" that achieves even better regret for the self-bounding analysis which is $O(\log T + \sqrt{C\log T/C} )$.

Taking this point to account, Zimmert et al. (2019) algorithm in stochastic and stochastic with adversarial corruption regimes has the same performance as Algorithm 1 in the paper. On the other hand their algorithm in adversarial setting has tight regret bound while Algorithm 1 has additional multiplicative factor $\sqrt{\log T}$ in the regret. Moreover the results on size-invariant is almost the same as Zimmert et al. (2019). So the only improvement is the data dependent bound for the adversary regime.


Still I consider this improvement as a good contribution as long as I can be convinced that the Zimmert et al. (2019)  is __not__ able to attain such a bound in _easy_ environments. To be more precise I would like to see a good discussion and also experimental results to support the inferior performance of Zimmert et al. (2019) algorithm in  _easy_ environments and obviously the superior performance of proposed algorithm.



__Quality:__
The proof generally follows self-bounding analysis by Zimmert and Seldin (2021) and optimistic predictors analysis by Wei and Luo (2018) and seems sound, however I did not track the full calculations in the supplementary materials.


## Linear Bandit Part of the Paper
The last section of the paper regarding the linear bandit seems incoherent to me since the major part of the paper is about how to obtain and analysis new algorithm by combing Zimmert et al., (2019) and optimistic loss predictors. I see that the authors aims obtain data dependent bound for linear bandit similar to semi-bandit, however, for the non-expert reader it is hard to follow this short section since no background and intuition about provided algorithms and regret bounds analysis is provided.

__Significance and Originality:__

The analysis of regret idea has no novelty since it is straightforward by just combining two existing bounds for regret. Achieving path-length regret bound for linear bandit is not significant since as the authors mentioned there are already two data dependent bounds for this problem based on _total quadratic variation of losses_ and _cumulative loss for optimal actions_ achieved by an algorithm provided by Ito et al., (2020). The worst case of these bounds are same as the path length regret bound, besides that there are rare cases that path length regret would benefit us over two former ones. So I consider this as a minor contribution.



__Clarity of the Paper:__
- Generally the paper is well written and easy to read. But the last section on Linear bandits seems irrelevant and so vague since there is no provided background, intuition, discussion and … about the subject. Theorem 3 in this section is vague and there is undefined variable in it and need to revised.



**Time Spent Reviewing:**

20

---

> ### Author Response · Authors · 2021-08-09
> **Response to Reviewer CeRv**
>
> First of all, we would like to express our gratitude to the reviewer for taking the time to read this paper and for the many helpful comments.
> We promise to use these feedbacks to improve the quality of our manuscript.
> We hope that the following responses address your concerns.
>
> > Result of the paper for stochastic with adversarial corruption is $O(\log T + \sqrt{C \log T})$ whereas the authors claim Zimmert et al., (2019), with some straightforward modification of their proof, achieve $O( \log T + C)$. By contrast with the same self-bounding analysis provided by Tsallis-INF paper (Zimmert and Seldin, 2021) , where this paper uses the same analysis approach, it is quite easy to show that Zimmert et al., (2019) algorithm in this intermediate regime obtains $O(\log T + \sqrt{C \log T})$.
>
> Now that we have been able to show that the algorithm by Zimmert et al. (2019) achieves $O (\log T + \sqrt{C \log T})$-regret,
> we will add this fact and its proof in the revised version.
> We believe that this fact is not entirely trivial as the regret bound by Zimmert et al. (2019) includes a term of $O(\sum_{t=1}^T \frac{1}{\sqrt{t}} \sum_{i:a_i^* = 1} (\gamma^{-1} - \gamma \log (1- x_{ti}))(1-x_{ti}))$ in contrast to Tsallis-INF by Zimmert and Seldin (2021).
> Due to this additional term, the self-bounding analysis in Zimmert and Seldin (2021) does not seem to be directly applicable, and at the time of submission we could not prove a $O(\log T + \sqrt{C \log T})$-bound.
>
>
> > Still I consider this improvement as a good contribution as long as I can be convinced that the Zimmert et al. (2019) is not able to attain such a bound in easy environments. To be more precise I would like to see a good discussion and also experimental results to support the inferior performance of Zimmert et al. (2019) algorithm in easy environments and obviously the superior performance of proposed algorithm.
>
> For now, we do not know if we can obtain data-dependent bounds by modifying the algorithm by Zimmert et al. (2019).
> If we straightforwardly combine their algorithm with the optimistic-prediction technique,
> we cannot use Lemma 5 in [Zimmert et al. (2019)] as the assumption of $-1 \leq \hat{\ell}_i$ in the lemma does not hold.
> Consequently,
> existing analysis methods for Tsallis entropy regularization do not directly apply.
>
>
> > In general there is a work by Masoudian and Seldin (2021) titled: "Improved Analysis of Robustness of the Tsallis-INF Algorithm to Adversarial Corruptions in Stochastic Multiarmed Bandits" that achieves even better regret for the self-bounding analysis which is
> $O(\log T + \sqrt{C \log T / C})$.
>
> Thanks for sharing this.
> We will add information on this literature in the revised version.
>
>
> > Achieving path-length regret bound for linear bandit is not significant since as the authors mentioned there are already two data dependent bounds for this problem based on total quadratic variation of losses and cumulative loss for optimal actions achieved by an algorithm provided by Ito et al., (2020). The worst case of these bounds are same as the path length regret bound, besides that there are rare cases that path length regret would benefit us over two former ones. So I consider this as a minor contribution.
>
> We believe that it is not uncommon to find situations where the path-length bounds would benefit us over the other data-dependent bounds.
> For example, if $\ell_{ti^*} = 0$ for $t \leq T / 2$ and $\ell_{ti^*} = 1$ for $t > T/2$,
> the path-length can be $O(1)$ while the total quadratic variation and the cumulative loss are $\Omega(T)$.
> A more realistic example would be situations in which $\ell_t$ (significantly) changes only $S (<< T)$ times in $T$ rounds,
> i.e.,
> $S := |\{ t \in [T-1] \mid \ell_t \neq \ell_{t+1} \} | << T$.
> Then the path-length is at most $O(d S)$ while the total quadratic variation and the cumulative loss can be $\Omega(T)$.
> Such a situation is expected to be common in practical applications.
> For example, in the application of the bandit shortest-path problem over road traffic networks, $\ell_t$ corresponds to the congestion level, which changes significantly only in rare cases, such as when new facilities are built.
>
> > But the last section on Linear bandits seems irrelevant and so vague since there is no provided background, intuition, discussion and ... about the subject. Theorem 3 in this section is vague and there is undefined variable in it and need to revised.
>
> The content of the last section is related to the other sections in the context of hybrid algorithms that exploit certain (unknown) tendencies of environments.
> As stated in the first sentence of the abstract and in the introduction,
> the fundamental aim of this paper is to break the limitations of worst-case analysis (e.g., $O(\sqrt{T})$-regret bounds) by
> developing algorithms that automatically exploit certain structures of environments to improve performance, without any prior knowledge
> regarding the environments.
> We position both best-of-three-worlds property and data-dependent bounds as examples of approaches to this end in this paper.
> Definitions of parameters $Q_q$ and $V_q$ are given in Table 3,
> and $F^*$ is a typo for $L^*$.
> In the revised version, we will add the definitions of these parameters immediately after Theorem 3 for readability.
>
>
> > I see that the authors aims obtain data dependent bound for linear bandit similar to semi-bandit, however, for the non-expert reader it is hard to follow this short section since no background and intuition about provided algorithms and regret bounds analysis is provided.
>
> In the revised version, we will add more explanation about the main subject, background and intuition regarding the results in this paper, especially for data-dependent regret bounds.
>
>
> > The paper in stochastic and stochastic with adversarial corruption regimes implicitly uses the uniqueness of the best action. This is needed for the self-bounding analysis and must be mentioned as a clear assumption.
>
> In the revised version, we will state this assumption more explicitly.
>
> > There is a small issue about using “best of three worlds” term. The best of one world result implies there is no better result than that and in other words there must be a matching lower bound for the result. However, obviously the stated result for stochastic with adversarial corruption is not minimax optimal (consider $C\geq T / log T$) so in this world the achieved bound seems a rather good bound but is not the best.
>
> Thanks for this comment.
> We will mention this fact in the revised version.

---

### Decision · Program_Chairs · 2021-09-27

**Decision:**

Accept (Poster)

**Comment:**

The reviewers agree that this is an interesting and significant contribution.